# CPEB4 is regulated during cell cycle by ERK2/Cdk1-mediated phosphorylation and its assembly into liquid-like droplets

Jordina Guillén-Boixet[1,2], Víctor Buzon[1,2], Xavier Salvatella[1,2,3], Raúl Méndez[1,2,3]*

[1]Institute for Research in Biomedicine, Barcelona, Spain; [2]The Barcelona Institute of Science and Technology, Barcelona, Spain; [3]Institució Catalana de Recerca i Estudis Avançats, Barcelona, Spain

**Abstract** The four members of the vertebrate CPEB family of RNA-binding proteins share similar RNA-binding domains by which they regulate the translation of CPE-containing mRNAs, thereby controlling cell cycle and differentiation or synaptic plasticity. However, the N-terminal domains of CPEBs are distinct and contain specific regulatory post-translational modifications that presumably differentially integrate extracellular signals. Here we show that CPEB4 activity is regulated by ERK2- and Cdk1-mediated hyperphosphorylation. These phosphorylation events additively activate CPEB4 in M-phase by maintaining it in its monomeric state. In contrast, unphosphorylated CPEB4 phase separates into inactive, liquid-like droplets through its intrinsically disordered regions in the N-terminal domain. This dynamic and reversible regulation of CPEB4 is coordinated with that of CPEB1 through Cdk1, which inactivates CPEB1 while activating CPEB4, thereby integrating phase-specific signal transduction pathways to regulate cell cycle progression.

*For correspondence: raul.mendez@irbbarcelona.org

**Competing interests:** The authors declare that no competing interests exist.

## Introduction

The vertebrate CPE-binding protein (CPEB) family of RNA-binding proteins is composed of four members, with CPEBs 2–4 being more closely related to each other than to CPEB1 (reviewed in refs. [*Fernandez-Miranda and Mendez, 2012*; *Ivshina et al., 2014*]). All four CPEBs share similar C-terminal RNA-binding domains, composed of two RNA recognition motifs (RRMs) in tandem, followed by a ZZ domain, which is a zinc-binding domain with a cross-braced zinc binding topology. Consequently, the different CPEBs recognize similar *cis*-acting cytoplasmic polyadenylation elements (CPEs) on target mRNAs (*Afroz et al., 2014*; *Igea and Méndez, 2010*; *Merkel et al., 2013*; *Novoa et al., 2010*; *Ortiz-Zapater et al., 2011*; *Pavlopoulos et al., 2011*). Although up to 20–30% of vertebrate genes are regulated by CPEs, they are not all activated simultaneously. Rather, the CPEs define a combinatorial code, based on their number and distance to the polyadenylation signal (PAS), as well as the presence of additional *cis*-acting elements, to determine the transcript-specific spatiotemporal translation pattern (*Belloc and Mendez, 2008*; *Eliscovich et al., 2008*; *Mendez et al., 2002*; *Pique et al., 2008*). The CPE-mediated spatiotemporal patterns of translational control are further determined by the differential regulation of the CPEBs. All four CPEBs have dual functions as translational repressors and translational activators (by promoting cytoplasmic polyadenylation). This duality is differentially regulated for individual CPEB family members through post-translational modifications. Thus, it is plausible that distinct regulatory pathways provide a platform to support translational activation by cytoplasmic polyadenylation in different biological scenarios. At least for CPEB1 and CPEB3, the regulatory elements are localized in the N-terminal domain (NTD), which is highly variable both in length and composition across various CPEB orthologs and paralogs (*Wang and Cooper, 2010*). Notably, CPEBs' NTDs contain intrinsically disordered regions

(IDR), which are more extended in CPEB2, CPEB3 and CPEB4. IDR are particularly prevalent in RNA-binding proteins and, over the last years, have been shown to drive the formation of dynamic membraneless organelles (such as P-bodies, germ granules, stress granules and the nucleolus) through phase separation (*Brangwynne et al., 2009*, *2011*; *Li et al., 2012*; *Castello et al., 2012*; *Lin et al., 2015*; *Patel et al., 2015*; *Nott et al., 2015*; *Molliex et al., 2015*).

For CPEB3 and its *Drosophila* and *Aplysia* orthologs, Orb2 and ApCPEB respectively, the switch from repressor to activator has been proposed to be regulated through a prion-like multimerization mechanism in neurons (*Kruttner et al., 2012*; *Majumdar et al., 2012*; *Raveendra et al., 2013*). These CPEBs contain polyglutamine stretches in their NTDs that promote the formation of amyloid-like fibrils. CPEB3 multimerization is regulated by non-proteolytic mono-ubiquitination and SUMOylation, and the aggregated form promotes the polyadenylation and translational activation of its target mRNAs (*Drisaldi et al., 2015*; *Khan et al., 2015*; *Pavlopoulos et al., 2011*).

CPEB1 is regulated by two sequential phosphorylation events during the first meiotic division. In prophase I (PI)–arrested *Xenopus* oocytes, unphosphorylated CPEB1 recruits the deadenylase poly (A)-specific ribonuclease (PARN) to inhibit translation (*Kim and Richter, 2006*). Upon progesterone stimulation, Aurora kinase A (AurKA) phosphorylates CPEB1 at S174, which remodels CPEB1-mRNP from a repressor to an activator complex, thereby inducing the early wave of cytoplasmic polyadenylation required for the activation of cyclin-dependent kinase 1 (Cdk1) and MI entry (*Barnard et al., 2004*; *Mendez et al., 2000a*, *2000b*; *Pique et al., 2008*). Once the oocytes reach metaphase I (MI), Cdk1 and Polo-like kinase 1 (Plk1) target CPEB1 to ubiquitin-mediated degradation, and this is required to activate the second or 'late' wave of polyadenylation and the MI–MII transition (*Mendez et al., 2002*; *Pique et al., 2008*; *Setoyama et al., 2007*). In the second meiotic division, a third or 'late-late' wave of polyadenylation is supported by CPEB4, which is synthesized during the early and CPEB1-mediated wave of polyadenylation (*Belloc and Mendez, 2008*; *Igea and Méndez, 2010*). Thus, CPEB4 is only present in the second meiotic division. However, the synthesis of CPEB4 alone is not sufficient to promote cytoplasmic polyadenylation, and progesterone stimulation is still required (*Novoa et al., 2010*). These observations suggest that CPEB4 is not constitutively active but has to be post-translationally modified to become active. Unlike CPEB1, CPEB4 does not harbour a consensus AurKA phosphorylation site; therefore, it is most likely activated by distinct phosphorylation event(s) during later meiotic phases. Hence, we sought to elucidate the post-translational regulation of CPEB4 that is required for it to support cytoplasmic polyadenylation after CPEB1 has been degraded and to drive the second meiotic division. Unravelling how CPEB4 activity is regulated will contribute to the overall understanding of the CPEB network in mRNA translational control and to defining the phase-specific contributions of individual CPEBs to the unidirectional progression through meiotic and mitotic cell cycle (*Giangarra et al., 2015*; *Igea and Méndez, 2010*).

Here we show that CPEB4 activity is regulated by ERK2- and Cdk1-mediated phosphorylation at 12 residues surrounding two intrinsically disordered regions (IDRs) in CPEB4's NTD. These phosphorylation events additively activate CPEB4 for cytoplasmic polyadenylation and, consequently, are required for proper meiotic progression. Furthermore, CPEB4 hyperphosphorylation keeps CPEB4 in a monomeric and active state. In contrast, unphosphorylated CPEB4 sequesters CPE-containing mRNAs into inactive, liquid-like droplets formed through intermolecular interactions between residues in CPEB4's NTD. These results illustrate how phase transitions can be regulated through post-translational modifications and explain how CPEB-specific mechanisms of regulation sustain and coordinate cytoplasmic polyadenylation during cell cycle progression, adapting to the cell signalling transduction pathways activated at each cell cycle phase.

## Results

### xCPEB4 is phosphorylated in 12 residues by ERK2 and Cdk1

To address whether *Xenopus* CPEB4 (xCPEB4) is post-translationally modified by phosphorylation or by alternative modifications in a cell cycle–dependent manner, we first microinjected mRNA coding for HA-tagged xCPEB4 in stage VI–arrested oocytes and then triggered meiotic resumption with progesterone. Progression from the PI arrest (stage VI) to MI (shown as germinal vesicle breakdown, GVBD) caused a change in mobility, which was maintained through anaphase/interkinesis (GVBD + 1

hr) and in the MII arrest (MII). This mobility shift was abrogated by treatment with lambda-phosphatase, indicating that xCPEB4 was phosphorylated in response to progesterone (*Figure 1B*). To further define the xCPEB4 region(s) phosphorylated in a cell-specific manner, we subdivided xCPEB4 into four fragments, with three (1–3) comprising sections of its NTD and the fourth comprising the whole RNA-binding domain (RBD) (*Figure 1A*). *In vitro* phosphorylation of full-length (FL) xCPEB4 or the four xCPEB4 fragments with extracts from MII oocytes indicated that multiple residues were targeted in fragments 1–3 but none, or only very few, in the RBD fragment (*Figure 1C*; *Figure 1—figure supplement 1A*). To further define the phase specificity of the phosphorylation events, we tested each fragment for *in vitro* phosphorylation with extracts from oocytes collected either prior to progesterone stimulation (PI arrested; VI) or at various times after GBVD, corresponding to MI (GBVD and +30 min), anaphase/interkinesis (+60 min and +90 min) and MII-arrest (+120 min and +150 min) (*Figure 1D*; *Figure 1—figure supplement 1B*). Neither FL xCPEB4 nor any of the fragments were phosphorylated by PI-arrested oocyte extracts. FL xCPEB4 and fragment 1 were phosphorylated in MI and remained phosphorylated throughout the subsequent cell cycle phases. In contrast, fragment 3 was differentially phosphorylated by the extracts from the distinct meiotic phases, following the characteristic kinetics of Cdk1 (*Mendez et al., 2002*), as shown by the phosphorylation of histone H1. The phosphorylation kinetics of fragment 2 was intermediate between fragments 1 and 3. In sum, these results indicate that xCPEB4 is phosphorylated at multiple residues in its NTD by distinct kinases that are differentially active during meiotic progression, one of which is most likely Cdk1.

To further determine the kinases targeting xCPEB4, we performed *in vitro* phosphorylation assays on FL-xCPEB4 and on fragments 1–3 using MII oocyte extracts as the kinase source, in the presence or absence of inhibitors of Cdks (roscovitine), p90Rsk (SL0101), Plk1 (BI-2536), MEK (U0126) or ERK (FR180204) (*Figure 2A,B*; *Figure 2—figure supplement 1*). All these kinases are active in MII (*Tunquist and Maller, 2003*) (for Cdks and ERK, Cdk1 and ERK2 are the active kinases in MI and MII). Phosphorylation of the FL protein was strongly inhibited by FR180204, partially inhibited by roscovitine and BI-2536, and unaffected by SL0101 and U0126, suggesting that ERK2, Cdk1 and Plk1, but not MEK or p90Rsk, target xCPEB4. Phosphorylation of fragments 1 and 2 was specifically inhibited by FR180204, whereas that of fragment 3 was more sensitive to roscovitine, suggesting that ERK2 and Cdk1 target specific regions of the xCPEB4 NTD. We further confirmed the role of ERK2 and Cdk1 in xCPEB4 phosphorylation by immunodepleting the kinases from phase-specific meiotic extracts, with the subsequent reduction of xCPEB4 phosphorylation (*Figure 2—figure supplement 2*). To determine the number of phosphorylated residues in each fragment, we phosphorylated the fragments 1–3 *in vitro* with extracts from oocytes collected at MI (GVBD), interkinesis (+1 hr) or MII arrest. The phosphorylated fragments were purified, digested and analysed by 2D phosphopeptide mapping (*Figure 2—figure supplement 3*). Fragment 1 yielded four main phosphopeptides (pp 1–4), with a weaker one detectable only after a long exposure (*Figure 2—figure supplement 4*, pp 5). Fragment 2 contained a single phosphopeptide (pp 6), while fragment 3 had six additional phosphopeptides (pp 7–12). These phosphopeptide patterns were qualitatively maintained from MI to MII (*Figure 2—figure supplement 3*). We then compared these 2D phosphopeptide maps with those obtained using recombinant ERK2 or Cdk1/cyclin B (*Figure 2C*). ERK2 generated pp 1–6 and pp 10, whereas Cdk1/cyclin B produced pp 1–3 (albeit with low efficiency), 8, 9 and 11–12. In addition, both purified kinases generated other phosphopeptides not observed after phosphorylation with oocyte extract (marked with an asterisk). The migration of some of the phosphopeptides was consistent with phosphorylation at multiple residues in the same peptide. To determine the identity of these residues, we analysed by mass spectrometry either FL xCPEB4 or N-terminal fragments that had been phosphorylated *in vitro* with MII oocyte extracts or recombinant ERK2 or Cdk1/cyclin B. A total of twelve phosphorylation sites in the NTD of xCPEB4 were identified, all of which were evolutionarily conserved (*Figure 2D*). Importantly, xCPEB4 phosphosites S97, S250, S253, T324, S328 and S330 were previously identified as phosphorylated residues in CPEB4 mammalian orthologs by high-throughput mass spectrometry screenings (PhosphoSitePlus; http://www.phosphosite.org).

To confirm that the phosphopeptides detected by 2D phosphopeptide mapping corresponded to the identified phosphosites, we generated a mutant version of the three N-terminal fragments by converting the serine/threonine residue to aspartic acid at the indicated sites, producing fragment 1 (S18/38/40/97D), fragment 2 (S250/253D) and fragment 3 (T324/S328/330/351/357/362D). These

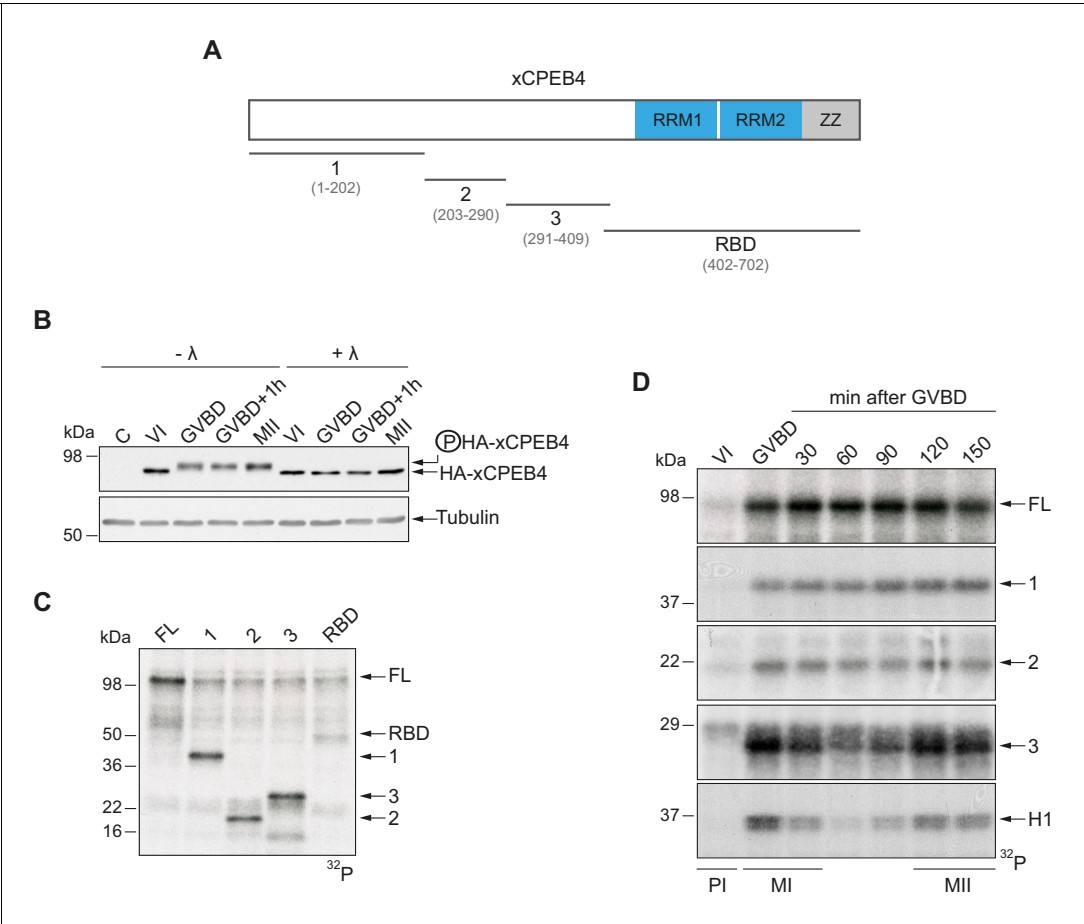

**Figure 1.** xCPEB4 NTD is phosphorylated in response to progesterone. (A) Schematic representation of xCPEB4 protein. The four protein fragments generated are: 1, from amino acid (aa) 1 to 202; 2, from aa 203 to 290; 3, from aa 291 to 409; and RBD, from aa 402 to 702. RRMs are shown in blue and the ZZ domain, in grey. (B) Lambda-phosphatase assay (λ) of oocytes overexpressing HA-xCPEB4 collected at the indicated times (VI, prophase I; GVBD, germinal vesicle breakdown; MII, metaphase II). Western blots with anti-HA antibody and anti-tubulin (as a loading control) are shown. C, control, corresponds to non-injected oocytes. (C) *In vitro* kinase assay of recombinant xCPEB4, full-length (FL) or the fragments 1, 2, 3 or RBD, using metaphase II oocyte extracts as the source of kinases (autoradiography, $^{32}$P). (D) Time course of the *in vitro* kinase assay of recombinant xCPEB4, FL or fragments (1, 2 and 3), using oocyte extracts collected at the indicated times (autoradiography, $^{32}$P). Histone H1 phosphorylation was used to follow Cdk1 activity. Three independent biological replicates were performed, with equivalent findings each time. See also *Figure 1—figure supplement 1*.

The following figure supplement is available for figure 1:

**Figure supplement 1.** Loading controls for *in vitro* kinase assays.

mutated fragments were phosphorylated with MII extracts or recombinant ERK2 or Cdk1/cyclin B (*Figure 2—figure supplement 4A,B*). Based on the resulting 2D phosphopeptide maps, we determined that most of the xCPEB4 phosphorylation sites had been identified. However, the mutated xCPEB4 fragments still contained three phosphorylation sites, corresponding to pp 1, 2 and 7, that were not detected in the mass spectrometry analysis. Accordingly, mutation of the 12 identified residues to alanine (12A) did not abolished the slower migration of xCPEB4 expressed in MII oocytes (*Figure 2—figure supplement 4C*). Thus, we conclude that xCPEB4 is phosphorylated in at least 12 sites, which most likely correspond to ERK2-mediated (S18/38/40/97/250/253/351) and Cdk1-mediated (T324/S328/330/357/362) target residues. Most of the residues we identified are in regions of the NTD predicted to have relatively low disorder (*Figure 2D*), as determined by PONDR VL-TX (Predictors of Naturally Disordered Regions; http://www.pondr.com).

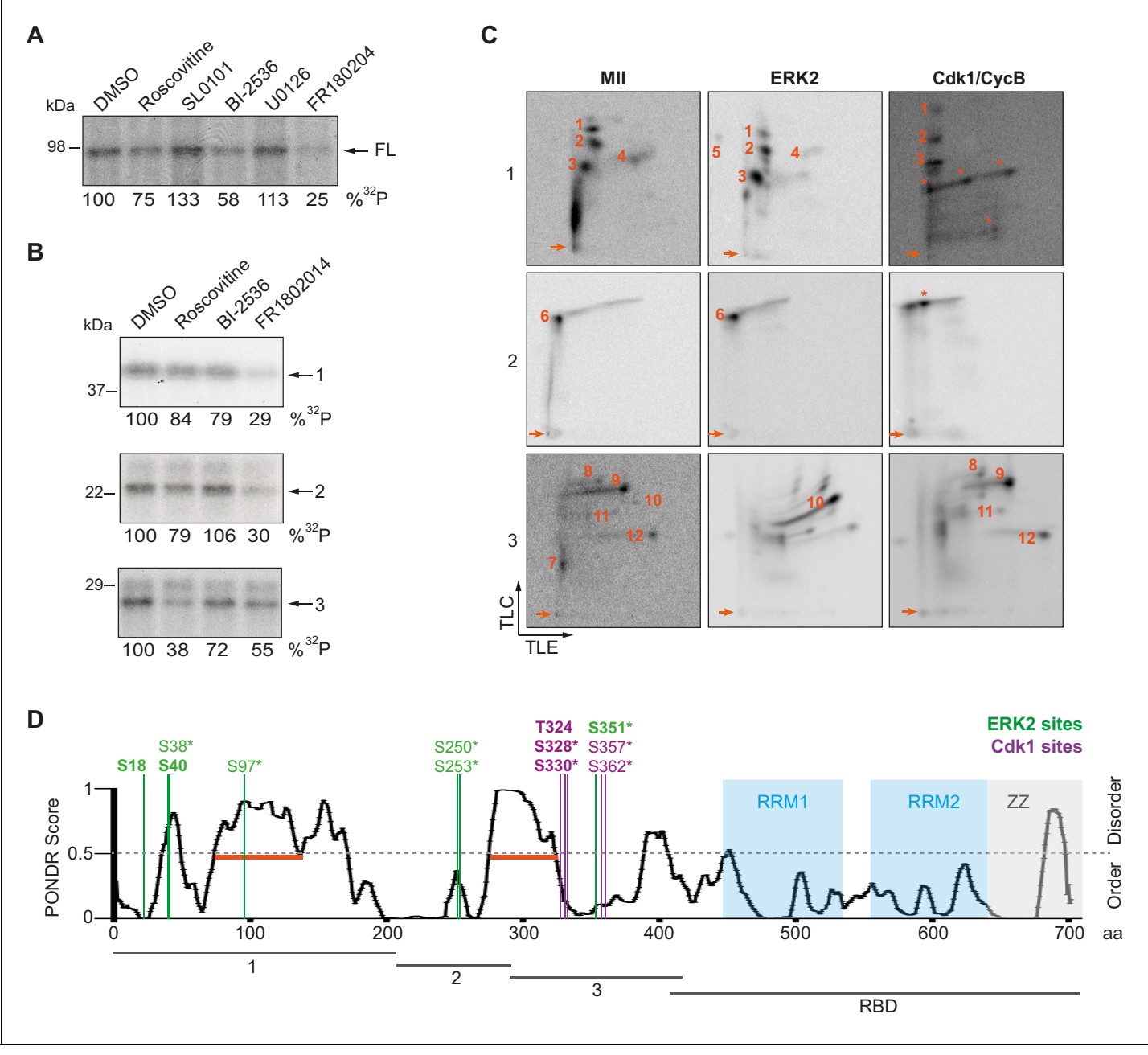

**Figure 2.** ERK2 and Cdk1 phosphorylate the xCPEB4 NTD. (A, B) *In vitro* kinase assay of (A) recombinant FL xCPEB4 or (B) fragments (1, 2 and 3) with metaphase II oocyte extracts treated with specific kinase inhibitors (roscovitine, Cdk inhibitor; SL0101, p90Rsk inhibitor; BI-2536, Plk1 inhibitor; U0126, MEK inhibitor; and FR180204, ERK inhibitor). DMSO was used as a negative control. The percentage of phosphorylation compared to DMSO is indicated (autoradiography, $^{32}$P). A representative experiment from three independent biological replicates is shown. See also *Figure 2—figure supplement 1–2*. (C) Two-dimensional phosphopeptide maps of xCPEB4 fragments (1, 2 and 3) phosphorylated with metaphase II (MII) oocyte extracts or with recombinant ERK2 or Cdk1/cyclin B. Phosphopeptides were resolved by thin-layer electrophoresis (TLE) followed by thin-layer chromatography (TLC). Arrows indicate sample origin. Phosphopeptides detected in MII were numbered. Asterisks (*) indicate phosphopeptides generated with recombinant kinases not present in MII. A representative experiment from three independent biological replicates is shown. See also *Figure 2—figure supplement 3*. (D) Disorder tendency (PONDR VL-TX predictor) and mass spectrometry phosphorylation site identification of xCPEB4. Asterisks (*) indicate phosphosites identified with MII extracts. Bold letters indicate phosphosites identified with either ERK2 or Cdk1/cyclin B. Green indicates ERK2 phosphorylation sites, while purple indicates Cdk1 phosphorylation sites. Red lines represent large disordered regions. The xCPEB4 fragments used are outlined. See also *Figure 2—figure supplement 4*.

The following figure supplements are available for figure 2:

*Figure 2 continued on next page*

*Figure 2 continued*

**Figure supplement 1.** Loading controls for *in vitro* kinase assays with inhibitors.

**Figure supplement 2.** *In vitro* kinase assay with Cdk1 and ERK2 immunodepleted extracts.

**Figure supplement 3.** Two-dimensional phosphopeptide maps kinetic.

**Figure supplement 4.** Confirmation of detected phosphosites.

## xCPEB4 phosphorylation is required for cytoplasmic polyadenylation and meiotic progression

We next tested whether these phosphorylation events are required for xCPEB4 activity in promoting cytoplasmic polyadenylation during the second meiotic progression.

First, we analysed the capacity of phospho-mimetic or phospho-null variants to compete or substitute endogenous xCPEB4. For this purpose, we followed the polyadenylation of a radioactive *Emi2* 3′-UTR probe microinjected into oocytes. Due to its arrangement of CPEs and the presence of AU-rich elements (AREs), *Emi2* mRNA is polyadenylated in the third (late-late) wave of cytoplasmic polyadenylation that takes place in interkinesis and MII (*Belloc and Mendez, 2008*; *Pique et al., 2008*) and which is supported by xCPEB4 (*Igea and Méndez, 2010*). In this assay, overexpression of active or activatable xCPEB4 functionally replaces the endogenous xCPEB4 and therefore sustains *Emi2* mRNA polyadenylation, whereas a non-activatable CPEB4 variant reaches a competitive equilibrium with the endogenous protein, shortening the poly(A) tail of the target mRNA (*Afroz et al., 2014*) (*Figure 3—figure supplement 1*). Mutation of all 12 xCPEB4 phosphorylated residues to alanine (12A) not only inactivated the protein, abolishing its capacity to support cytoplasmic polyadenylation (*Figure 3A,B*) but was also a better competitor of polyadenylation than a truncated variant of xCPEB4 lacking the NTD (*Figure 3C*), suggesting that the 12A mutant promotes poly(A) tail shortening. The mutant 12Am, an xCPEB4 phospho-null mutant that additionally cannot bind RNA (with mutations at Y490A and K595A), did not influence competition for polyadenylation (*Afroz et al., 2014*) (*Figure 3A,B*), indicating that the competition effect requires RNA binding. In contrast, a phospho-mimetic xCPEB4 variant (12D), in which all the phosphosites were mutated to aspartic acid, was even less competitive than the WT xCPEB4 (*Figure 3A,B*; *Figure 3—figure supplement 2*), suggesting that this mutant is constitutively active. xCPEB4 mutants in which only the seven ERK2 phosphorylation sites (7A) or the five Cdk1 phosphorylation sites (5A) were mutated to alanine showed partial and additive effects, indicating that these two kinases contribute equally to the activation of xCPEB4 (*Figure 3D,E*). Other partial mutations (10A and 8A; *Figure 3D,E*; *Figure 3—figure supplement 2*) also showed a partial and additive effect for the combined accumulation of mutations, indicating that all 12 phosphorylation events, mediated by the two kinases, are equally important to fully activate xCPEB4, and that a minimum of 10 phosphosites are required. Compared to 12Am, the wild-type CPEB4 (WT) was slightly competitive with endogenous CPEB4 protein, most likely due to inefficient activation or recruitment of co-factors. For this experiment, xCPEB4 variants were expressed from an mRNA where CPEB4-3′UTR was substituted by a non-CPE regulated 3′-UTR, allowing the mutant expression to scape the cell cycle regulation of endogenous CPEB4.

Second, we explored the capacity of various xCPEB4 phosphorylation mutants to compensate for the depletion of endogenous xCPEB4 in the progression of meiosis to MII arrest, as well as in the polyadenylation of an endogenous MII target mRNA (*Emi2*). Oocytes depleted of xCPEB4 completed the first meiotic division normally but failed to properly assemble chromosomes into a single metaphase plate in MII, displaying scattered chromosomes as compared to control oocytes (*Figure 3F*). Nevertheless, xCPEB4-depleted oocytes were able to properly extrude the polar body, indicating that the first meiotic division was completed successfully. Furthermore, xCPEB4 depleted oocytes failed to properly polyadenylate *Emi2* mRNA in the second meiotic division (*Figure 3G*). We next overexpressed xCPEB4, using WT and phosphorylation mutants under the control of xCPEB4 3′-UTR, in xCPEB4-depleted oocytes. The WT and 12D xCPEB4 forms, but not the 12A mutant, rescued chromosomal assembly in the MII metaphase plate, as well as *Emi2* polyadenylation

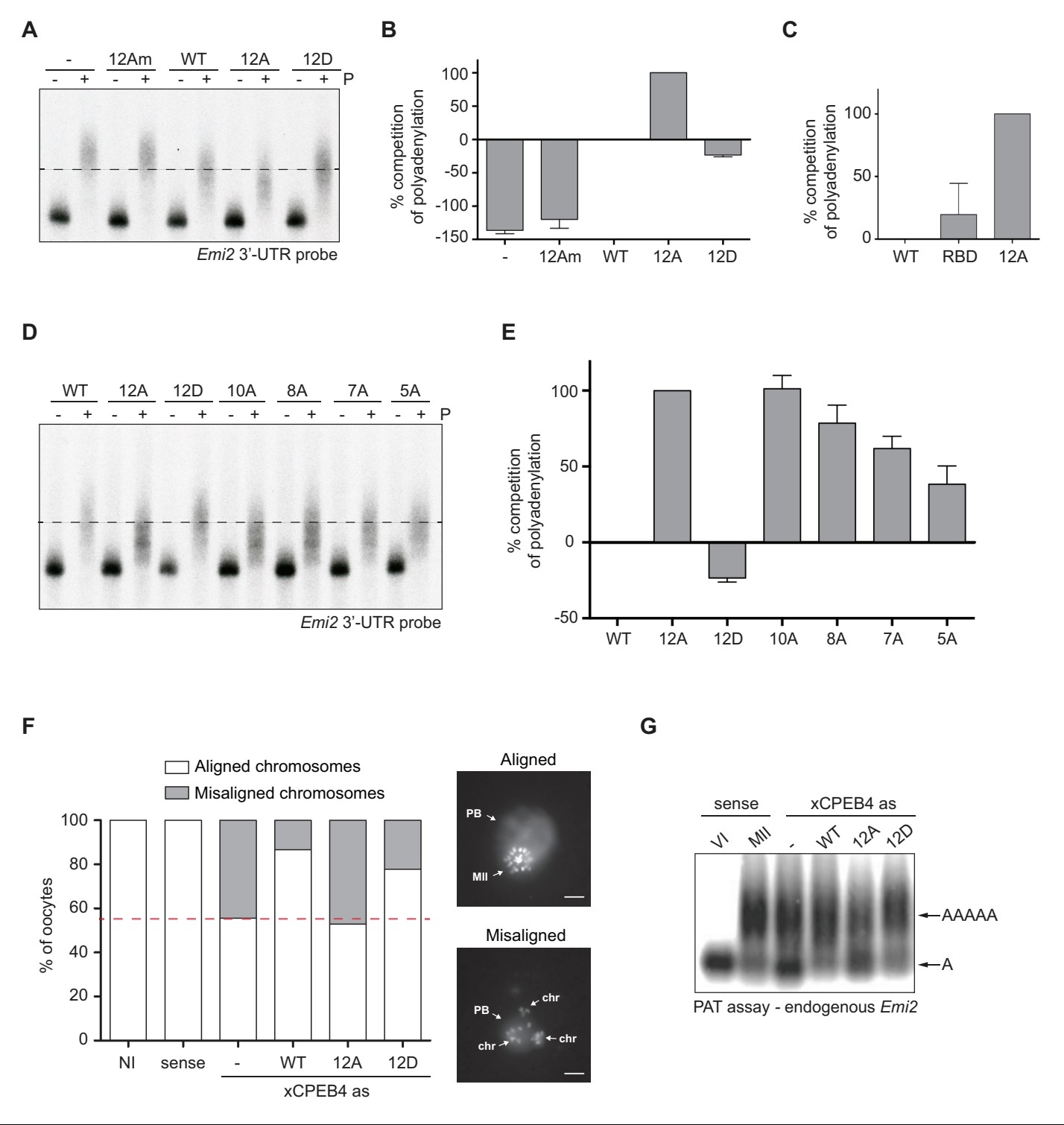

**Figure 3.** xCPEB4 hyperphosphorylation is required for cytoplasmic polyadenylation and meiotic progression. (A–E) *In vivo* competition assays. (A, D) *Emi2* 3'-UTR radioactive probe polyadenylation in the absence (–) or presence (+) of progesterone (P) in oocytes overexpressing wild-type (WT) HA-xCPEB4 or the following mutants: 12A or 12D, which have the 12 phosphosites mutated to alanine or aspartic acid, respectively; 12Am, 12A mutant with two additional mutations that disrupt RNA binding in the RBD; 10A, with all the identified phosphosites except S18 and S40 mutated to alanine; 8A, with S250/253/T324/S328/330/351/357/362 mutated to alanine; 7A, with ERK2 phosphosites mutated to alanine; 5A, with Cdk1 phosphorylation sites mutated to alanine. The dashed line marks the median polyadenylation in the WT. (B, C, E) Percentage of competition of polyadenylation calculated from three independent biological replicates. Results are shown as means and SEM. See also *Figure 3—figure supplement 1–2*. (F, G) Rescue

*Figure 3 continued on next page*

*Figure 3 continued*

experiment. Oocytes were injected with xCPEB4 sense or antisense (as) oligonucleotides (NI, non-injected oocytes). After 16 hr, oocytes were injected with mRNAs encoding WT HA-xCPEB4 or the 12A or 12D phosphorylation mutants. (F) Oocytes were collected at metaphase II, fixed and stained with Hoechst. A representative image of each phenotype observed, with aligned or misaligned chromosomes, is shown (PB, first polar body; MII, second metaphase plate; chr, misaligned chromosomes). The percentage of oocytes with each phenotype is plotted in the graph (n = 10). The red line marks the percentage of oocytes with aligned chromosomes in the antisense condition. Scale bar, 10 μm. A representative experiment from three independent biological replicates is shown. (G) RNA from metaphase II oocytes was extracted and the polyadenylation status of endogenous *Emi2* mRNA was measured by Poly(A) tail (PAT) assay followed by Southern blot. See also *Figure 3—figure supplement 3*, *Figure 3—figure supplement 4*.

The following figure supplements are available for figure 3:

**Figure supplement 1.** Experimental design and analysis of the competition experiment.

**Figure supplement 2.** Expression levels of xCPEB4 variants used in competition experiments.

**Figure supplement 3.** *In vivo* rescue experiment.

**Figure supplement 4.** xCPEB4 does not differentially interact with cofactors depending on its phosphorylation status.

(*Figure 3F,G*; *Figure 3—figure supplement 3B*). Consistent with the observation that the 5A and 7A mutants were partially competitive for polyadenylation (see *Figure 3E*), only the fully phosphorylated xCPEB4 was able to rescue the MII meiotic arrest (*Figure 3—figure supplement 3*). Thus, we conclude that ERK2 and Cdk1 phosphorylate and activate xCPEB4 in the 12 identified sites in order to sustain cytoplasmic polyadenylation in the second meiotic division.

## xCPEB4 phosphorylation regulates its capacity to phase separate into liquid-like droplets

To define the functional consequence of xCPEB4 phosphorylation, we first analysed whether these phosphorylation events affect its capacity to recruit cofactors, as is the case for CPEB1, the phosphorylation of which by AurKA causes a switch of activity from recruiting the deadenylase PARN to assembling the CPSF/GLD2 polyadenylation complex (*Barnard et al., 2004*; *Kim and Richter, 2006*; *Mendez et al., 2000b*). We overexpressed HA-tagged 12A (phospho-null) and 12D (phospho-mimetic) xCPEB4 mutants in oocytes and, after progesterone stimulation, analysed the co-immuno-precipitated proteins by SDS silver staining and mass spectrometry. Most of the identified interacting proteins were enriched for both mutants, with some exceptions, such as chaperones (Hspd1) and RNA helicases (Ddx1 and Ddx20), which were present only in the 12A immunoprecipitation (*Figure 3—figure supplement 4A*; *Supplementary file 1*). We specifically tested whether the two mutants could still recruit GLD2, the cytoplasmic poly(A) polymerase directly responsible for poly(A) tail elongation. Again, we did not detect any differences between the phospho-mimetic and the phospho-null xCPEB4 mutants (*Figure 3—figure supplement 4B*).

We next addressed whether phosphorylation of xCPEB4 causes changes in its subcellular localization that could explain the differences in its activity. We overexpressed FL or NTD GFP-tagged xCPEB4 constructs, using either the wild-type or the 12A and 12D mutants for each, in U2OS cells. GFP and the GFP-xCPEB4 RNA-binding domain (RBD) were used as controls (*Figure 4A,B*; *Figure 4—figure supplement 1*). Both WT and 12A GFP-xCPEB4 phase-separated into cytoplasmic spherical granules when expressed in U2OS cells. In contrast, the 12D mutant exhibited diffuse cytoplasmic distribution (*Figure 4A,C*). The WT, 12A and 12D xCPEB4 NTDs expressed in U2OS cells distributed in patterns similar to their corresponding FL variants (*Figure 4B,D*). The controls, of either GFP alone or the GFP-xCPEB4 RBD, exhibited diffuse nuclear and cytoplasmic distribution, suggesting that the RBD does not contribute to the granular distribution of xCPEB4. The analysed cells showed equivalent fluorescence intensities (*Figure 4E,F*), implying that the differential ability to phase separate is not a consequence of distinct protein concentration. These results indicate that, when non-phosphorylated, the intrinsically disordered NTD of xCPEB4 phase separates into large assemblies in non-mitotic cells. Co-staining of P-bodies and stress granule markers (4E-T and DDX6; *Figure 4—figure supplement 2*) showed no clear co-localization of these with xCPEB4. Thus,

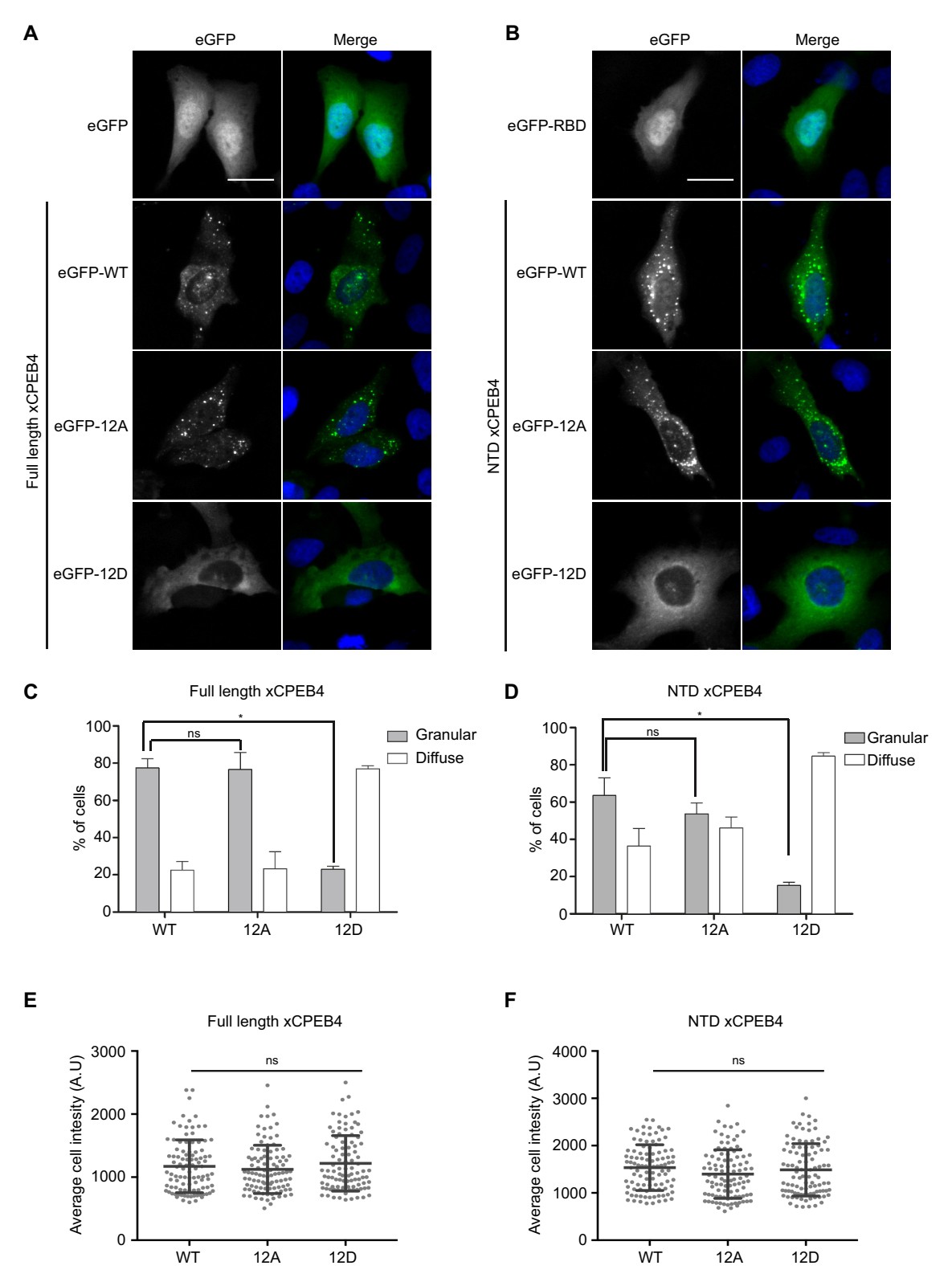

**Figure 4.** Non-phosphorylated NTD promotes xCPEB4 distribution in cytoplasmic granules. (**A, B**) Representative image of (**A**) full-length eGFP-xCPEB4 or (**B**) the N-terminal domain, using the wild-type (WT) or phosphorylation mutants 12A or 12D (with the 12 phosphorylation sites mutated to alanine or aspartic acid, respectively), transfected in U2OS cells. As controls, eGFP and eGFP-RBD (RNA-binding domain) were used. Merge images show eGFP in green and DAPI in blue. Scale bar, 25 μm. (**C, D**) Quantification of the different expression patterns observed in *Figure 4A* (**C**) and *Figure 4B* (**D**). 100

*Figure 4 continued on next page*

*Figure 4 continued*

cells were analysed and classified as having a granular (grey bars) or diffuse (white bars) expression pattern. The percentage of cells with each phenotype was calculated from three independent biological replicates. Results are shown as means and SEM. Significance was addressed with a Sidak's multiple comparisons test. *$p < 0.05$; ns, non-significant. (E, F) Average cell intensity ($n = 100$ cells) of the cells analysed in *Figure 4C* (E) and *Figure 4D* (F). Results are shown as means (SD). Significance was determined by the Student's t-test. *$p < 0.05$; ns, non-significant. See also *Figure 4— figure supplement 1–2*.

The following figure supplements are available for figure 4:

**Figure supplement 1.** Expression of eGFP-xCPEB4 variants.

**Figure supplement 2.** xCPEB4 cytoplasmic granules are distinct from P-bodies or stress granules.

although CPEB4 re-localizes to stress granules in response to arsenite (*Chang and Huang, 2014*), inactive xCPEB4 in non-stressed cells forms cytoplasmic granules that differ from P-bodies or stress granules. To better characterize the nature of these assemblies, we performed FRAP experiments with FL or NTD GFP-xCPEB4. We determined a half-time of maximum recovery (Half Max) of $1.8 \pm 0.11$ s for FL GFP-xCPEB4 and of $0.76 \pm 0.07$ s for the NTD (*Figure 5A*; *Figure 5—figure supplement 1*). These fast recovery rates indicate that the molecules forming the granules had a liquid-like behaviour and were in constant exchange with the surrounding cytoplasm. This feature, together with the spherical shape of the granules and the fact that we could observe fusion events over time (*Figure 5B*; *Video 1*), suggests that xCPEB4 is able to phase separate into liquid-like droplets. The difference in recovery between the FL and the NTD forms indicates that the RBD of xCPEB4, and most probably the bound mRNA, are not required for the self-assembly but influence the dynamics of these droplets.

We next sought to determine whether the phospho-dependent transition between the liquid-like droplet state and the monomeric active state of CPEB4 occurs endogenously. During meiosis, xCPEB4 accumulates in MII, during which ERK2 and Cdk1 are already active; consequently, xCPEB4 only exists in a phosphorylated active state. Thus, to evaluate the transition between the liquid-like droplet state and the monomeric active state of endogenous CPEB4, we studied human CPEB4 (hCPEB4) during the mitotic cell cycle in U2OS cells (hCPEB4 and xCPEB4 share 93% of identity and the identified phosphorylation sites are conserved). To this end, cells were synchronized in mitosis (with nocodazole or RO-3306) and selected by mitotic shake-off. As in meiosis, endogenous hCPEB4 presented a mobility shift in M-phase that disappeared upon lambda-phosphatase treatment, indicating that hCPEB4, like xCPEB4, is phosphorylated in M-phase (*Figure 6A*). To assess if this phosphorylation reflected the transit from the liquid-like droplet state to the monomeric state, we first detected endogenous hCPEB4, in mitotic and non-mitotic cells, by immunofluorescence. Even though it did appear that in mitotic cells the distribution of hCPEB4 was more diffuse than in G1 cells, the change in size/shape of mitotic cells, as well as the increase of hCPEB4 levels during mitosis, made it difficult to quantify the distribution of hCPEB4 along the cell cycle phases (*Figure 6—figure supplement 1*). To avoid these limitations, we analysed the formation of hCPEB4 assemblies by sedimentation in sucrose gradients. Vinculin was used as a reference control. 75% of unphosphorylated hCPEB4 from asynchronous cells was located in dense sucrose fractions (15–23), indicating that CPEB4 is forming large assemblies. In contrast, 84% of phosphorylated hCPEB4 from mitotic cells was located in lighter sucrose fractions (1–13), suggesting that it is in a monomeric state (*Figure 6B, C*; *Figure 6—figure supplement 2*). We next analysed the association of known CPEB1-co-factors (PARN, Symplekin, CPSF2 and GLD2) with phosphorylated/monomeric or unphosphorylated/phase-separated CPEB4. First, we tested the distribution of those cofactors along the sucrose gradients (*Figure 6B*). Symplekin, CPSF2 and GLD2 were found in both dense and light fractions, co-migrating with CPEB4 in both the liquid-like droplet state and the monomeric state. CPEB4 Co-IPs confirmed that Symplekin and CPSF2 interact with phosphorylated and unphosphorylated CPEB4 (*Figure 7A*), as did GLD2 with xCPEB4 12A and 12D mutants in the oocytes (*Figure 3—figure supplement 4*). These results are consistent with the association of Symplekin, CPSF and GLD2 to both repressor and activator CPEB1-complexes (*Barnard et al., 2004*; *Kim and Richter, 2006*). However, PARN distribution was restricted to light fractions (*Figure 6B*), indicating that PARN is not responsible of

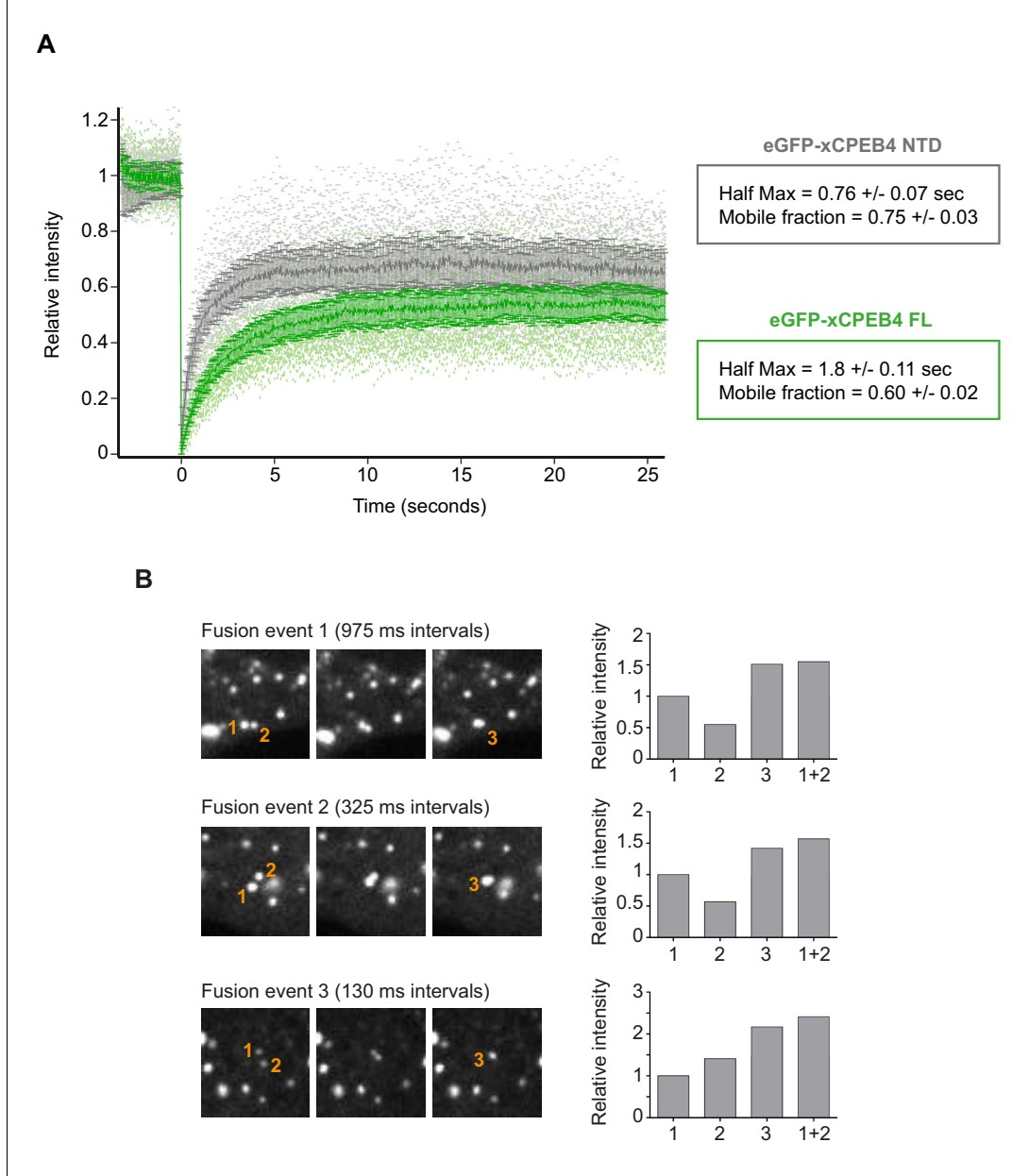

**Figure 5.** xCPEB4 cellular assemblies show liquid-like behaviours. (**A**) Fluorescence recovery after photobleaching (FRAP) of eGFP-xCPEB4 full-length (green) or the N-terminal domain (NTD, grey) transfected in U2OS cells. Raw data (dots) and average recovery curves (line) are shown. Error bars show 95% confidence interval. The half-time of maximum recovery (Half Max) and the mobile fraction are specified as means and SEM ($n = 20$). See also *Figure 5—figure supplement 1*. (**B**) Sequential images showing fusion events of xCPEB4 granules (left). The relative intensities of the fusing granules (1 and 2) and the resulting granule (3) are shown, as well as the expected intensity from the fusion of granules 1 and 2 (1 + 2) (right). See also **Video 1.**

The following figure supplement is available for figure 5:

**Figure supplement 1.** Fitting FRAP curves.

mRNA deadenylation in CPEB4 liquid-like droplets. We also analysed ribosome association (S6), which despite of being in the dense fractions of the gradients were not part of CPEB4 liquid-like droplets (*Figuers 6B* and *7A*). Thus, suggesting that CPEB4 liquid-like droplets are not active in translation. Altogether, these results are consistent with the mass spectrometry results

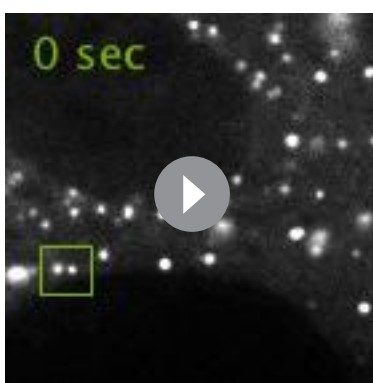

**Video 1.** Live imaging of U2OS cells transfected with GFP-xCPEB4 NTD. The fusion of two liquid-like droplets is shown inside the green box.

(*Supplementary file 1*), indicating that the change in CPEB4 activity upon phosphorylation are linked to its phase-transition and not to the differential recruitment of cofactors.

We then checked CPEB4 mRNA association in its two different states, phosphorylated/monomeric and unphosphorylated/phase-separated, analysing three known targets that are specifically polyadenylated and activated in mitosis and in a CPEB4 dependent manner: *Cdc20*, *Spop* and *Mnt* (*Novoa et al., 2010*). Thus, we performed RNA-co-immunoprecipitation of CPEB4 from light (5–9) and dense (17–21) sucrose gradient fractions in asynchronous and M-phase cells (*Figure 7B*). In asynchronous cells, 63% of *Cdc20*, 92% of *Spop* and 78% of *Mnt* mRNAs are bound by the CPEB4 in large-assemblies. In M-phase synchronized cells, 75% of *Cdc20*, 83% of *Spop* and 88% of *Mnt* mRNAs are bound by the free CPEB4 (*Figure 7B*). These results show that both states of CPEB4 are equally able to bind target mRNAs and that in asynchronous cells, the larger proportion of these CPEB4 targets are in the liquid-like droplets devoiced of ribosomes and, therefore, translationally inactive.

We concluded that, in interphase, unphosphorylated CPEB4 phase separates into liquid-like droplets that recruit mRNAs but are inactive for cytoplasmic polyadenylation and translation (*Giangarra et al., 2015*), whereas in M-phase, CPEB4 is phosphorylated by ERK2 and Cdk1 and recovers its monomeric form, which can drive the cytoplasmic polyadenylation of target mRNAs.

To further define whether CPEB4 assemblies correspond to self-association of CPEB4, we purified the NTD of WT, 12A, and 12D xCPEB4 variants and determined their intrinsic capacity to phase separate into liquid-like droplets *in vitro* by means of dynamic light scattering (DLS) and transmission electron microscopy (TEM). DLS measurements revealed that the WT and 12A NTDs had a hydrodynamic diameter > 1000 nm, while 12D presented a diameter of 10 nm (*Figure 8A*), which is in good agreement with that predicted for an intrinsically disordered protein of 459 residues. The hydrodynamic diameter > 1000 nm observed for the WT NTD, but not the equivalent 12A mutant, reverted to 10 nm when the protein was phosphorylated *in vitro* with ERK2 and Cdk1/cyclin B (*Figure 8B,C*). Thus, phosphorylation of the 12 identified residues by these two kinases controls the formation of large and dynamic self-assemblies through interactions of xCPEB4's N-terminal intrinsically disordered regions. When analysed by TEM, the unphosphorylated WT and the 12A xCPEB4 NTDs, but not the phosphorylated WT or the 12D NTDs, formed spherical structures that resemble liquid-like droplets (no fibers were detected in any case) (*Figure 8D*).

We next tested whether the FL protein within the liquid-like droplets could specifically bind CPE-containing RNA. We incubated recombinant xCPEB4 with radiolabelled cyclin B1–3'-UTR derived RNA-probes with (B1) or without (B1-123) CPEs (*Pique et al., 2008*). Then, this mixture was filtered through 0.22-μm-pore-size cellulose acetate filters, where 1 μm xCPEB4 liquid-like droplets, but not the soluble monomeric protein, were retained. Soluble xCPEB4, was recovered in the flow-through (*Figure 8E*). When tested for radiolabelled RNA binding, the membrane-bound FL-xCPEB4 was able to specifically retain the B1 RNA, but not the B1-123 (*Figure 8F,G*). This RNA-binding capacity of the liquid-like droplets required the RBD, as the NTD on its own did not retain the radiolabelled probe. Partial phosphorylation of FL-xCPEB4 resulted in a proportional release of phosphorylated xCPEB4 from the liquid-like droplets (*Figure 8E*) and a reduction of CPE-containing RNA probe retention in the filter (*Figure 8F,G*). Thus, the unphosphorylated NTD of CPEB4 promoted the self-assembly of liquid-like droplets, while the RBD recruited CPE-containing RNAs to this structures.

## Discussion

At the onset of the PI–MI meiotic transition, CPEB4 synthesis is activated by CPEB1-mediated cytoplasmic polyadenylation of its maternal mRNA (*Igea and Méndez, 2010*). However, CPEB4 has to

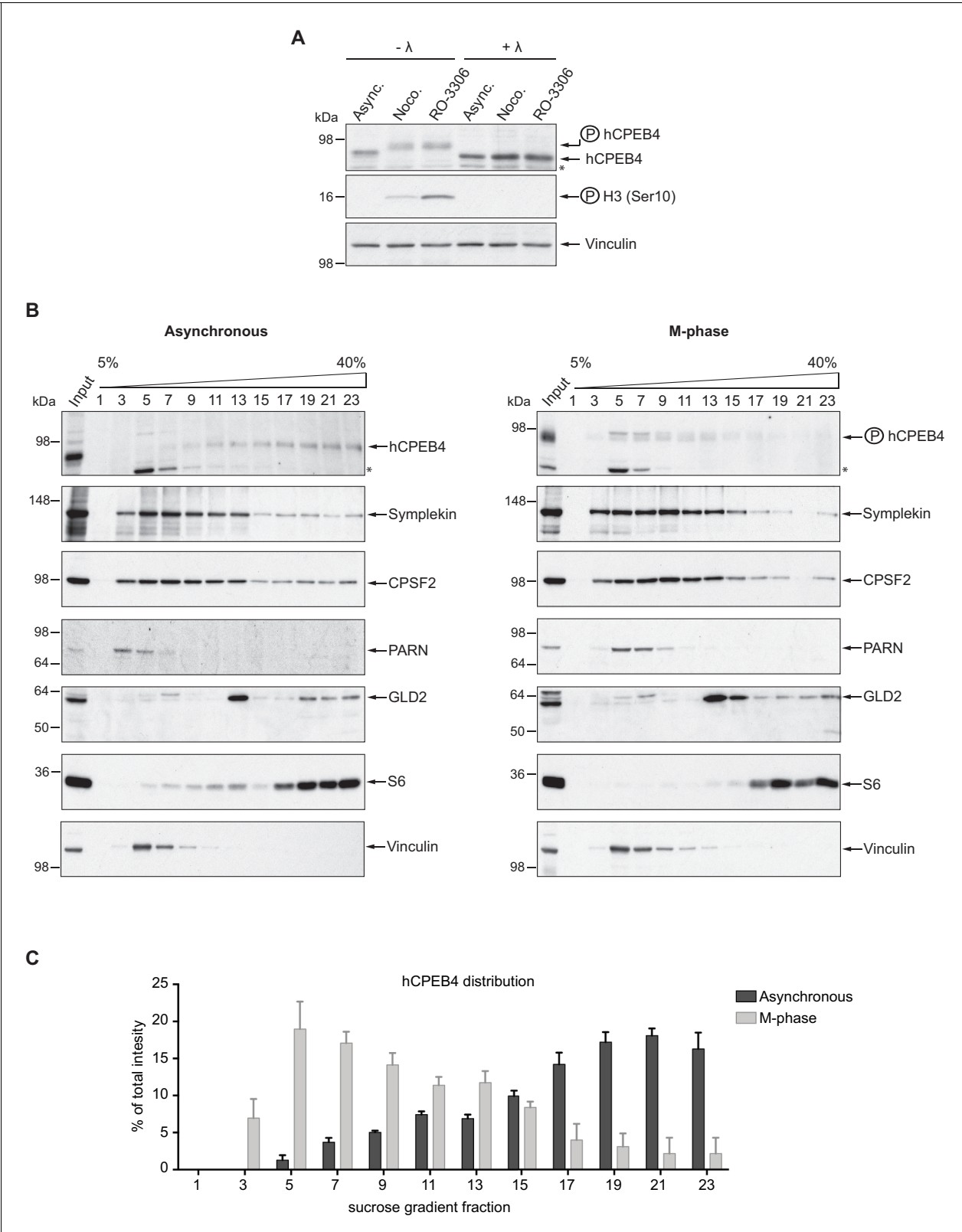

**Figure 6.** Endogenous hCPEB4 is phosphorylated and monomeric in M-phase. (**A**) Lambda-phosphatase assay (λ) of cell extracts from asynchronous (Async.) or M-phase U2OS cells synchronized with either nocodazole (Noco.) or RO-3306. Western blots against hCPEB4, the mitotic marker phospho-histone H3 and Vinculin (as a loading control) are shown. See also *Figure 6—figure supplement 1*. (**B**) Cell extracts from asynchronous or M-phase U2OS cells (synchronized with nocodazole) were resolved in 5–40% sucrose gradients. Distribution of hCPEB4, Symplekin, CPSF2, PARN, GLD2 and the

*Figure 6 continued on next page*

*Figure 6 continued*

ribosomal protein S6 along the gradient are shown. Vinculin was used as a reference control. Asterisks indicate unspecific bands. See also *Figure 6—figure supplement 2*. (C) Quantification of hCPEB4, from asynchronous and M-phase cells, distribution along 5–40% sucrose gradients. Results, from three independent biological replicates, are shown as means and SEM.

The following figure supplements are available for figure 6:

**Figure supplement 1.** Immunofluorescence of endogenous hCPEB4 in U2OS cells.

**Figure supplement 2.** hCPEB4 (from M-phase cells synchronized with RO-3306) distribution along 5–40% sucrose gradients.

be activated before it can sustain polyadenylation in the second meiotic division (*Novoa et al., 2010*). Our results indicate that CPEB4 activation is driven by ERK2- and Cdk1-mediated hyperphosphorylation of at least 12 residues in the intrinsically disordered NTD. Therefore, the sequential functions of CPEB1 and CPEB4 to drive unidirectional progression through the two meiotic divisions result from two redundant mechanisms that are both translational and post-translational (*Figure 9*). Thus, CPEB1 is activated by AurKA upon meiotic resumption from PI arrest (*Mendez et al., 2000a*), leading to the translation of the mRNAs encoding Mos, cyclin B1 and CPEB4 (*Igea and Méndez, 2010*; *Pique et al., 2008*). This in turn activates the kinases ERK2 and Cdk1, which phosphorylate CPEB4 and allow it to sustain cytoplasmic polyadenylation after MI, when AurKA is inactivated (*Pascreau et al., 2008*) and PP1 (which dephosphorylates CPEB1 S174) expression increases (*Belloc and Mendez, 2008*; *Tay et al., 2003*). Concomitantly, Cdk1 triggers the degradation of CPEB1 (*Mendez et al., 2002*), thus reinforcing the substitution of CPEB1 by CPEB4 for the second meiotic division (*Igea and Méndez, 2010*). Hence, CPEB1 and CPEB4 activities are coordinated, not only because CPEB1 controls the synthesis of CPEB4, but also because the same kinase that triggers the degradation of CPEB1—namely, Cdk1—activates CPEB4.

Unlike CPEB1, for which phosphorylation leads to the differential recruitment of cofactors, for CPEB4 we have not detected any specific cofactor that could explain the functional switch that CPEB4 experiences upon phosphorylation. Instead, CPEB4 phosphorylation controls its phase separation into liquid-like droplets, which are inactive for cytoplasmic polyadenylation and sequester CPE-containing mRNAs. Thence, our data support the idea that it is the phosphorylation-regulated phase-transition itself, and not the differential binding to positive or negative cofactors, what controls CPEB4 function. Interestingly, this correlation between phosphorylation and dissociation of the liquid-like droplets also occurs during the mitotic cell cycle, indicating that this regulation contributes to the CPEB4-dependent and M-phase–specific activation of CPE-regulated mRNAs (*Giangarra et al., 2015*; *Novoa et al., 2010*).

CPEB4 assemblies share the key features of non-membrane–bound organelles that behave like dynamic liquid droplets (*Weber and Brangwynne, 2012*). These dynamic, high-density liquid-like droplets are usually formed by repetitive, low-complexity intrinsically disordered regions, which self-assemble in micrometre-sized, near-spherical particles separated spontaneously from aqueous solutions (*Bah et al., 2015*; *Nott et al., 2015*). Hence, CPEB4 assemblies are structurally and functionally distinct from the more static and inheritable amyloid-like filaments formed by active CPEB3 multimers (*Han et al., 2012*; *Li et al., 2012*). Additionally, CPEB4 liquid-like droplets are reminiscent of other two-component models for mRNA recruitment to intracellular granules: RBPs bind to mRNA via an RBD and phase separate by the interaction of their low-complexity sequences (*Lin et al., 2015*; *Nott et al., 2015*; *Weber and Brangwynne, 2012*). The dynamism of these structures renders potential for regulated reversibility. In the case of CPEB4, its hyperphosphorylation results in the dissociation of the liquid-like droplets, most probably through the disruption of electrostatic interactions along its disordered NTD. This mechanism provides a novel example of how phase-transitions driven by disordered regions can be regulated post-translationally (*Nott et al., 2015*; *Han et al., 2012*), and unveils a new mechanism of translational regulation during cell cycle.

Given that CPEB4 has the potential to regulate up to 20% of the genome, with the flexibility of the high-specificity but low-affinity interactions promoted by the NTD and the hysteresis provided by the multisite phosphorylation regulation, it generates a system ideally suited to serve as a hub in scale-free networks (*Dunker et al., 2005*). Indeed, the regulation of CPEB4 by the additive effect of

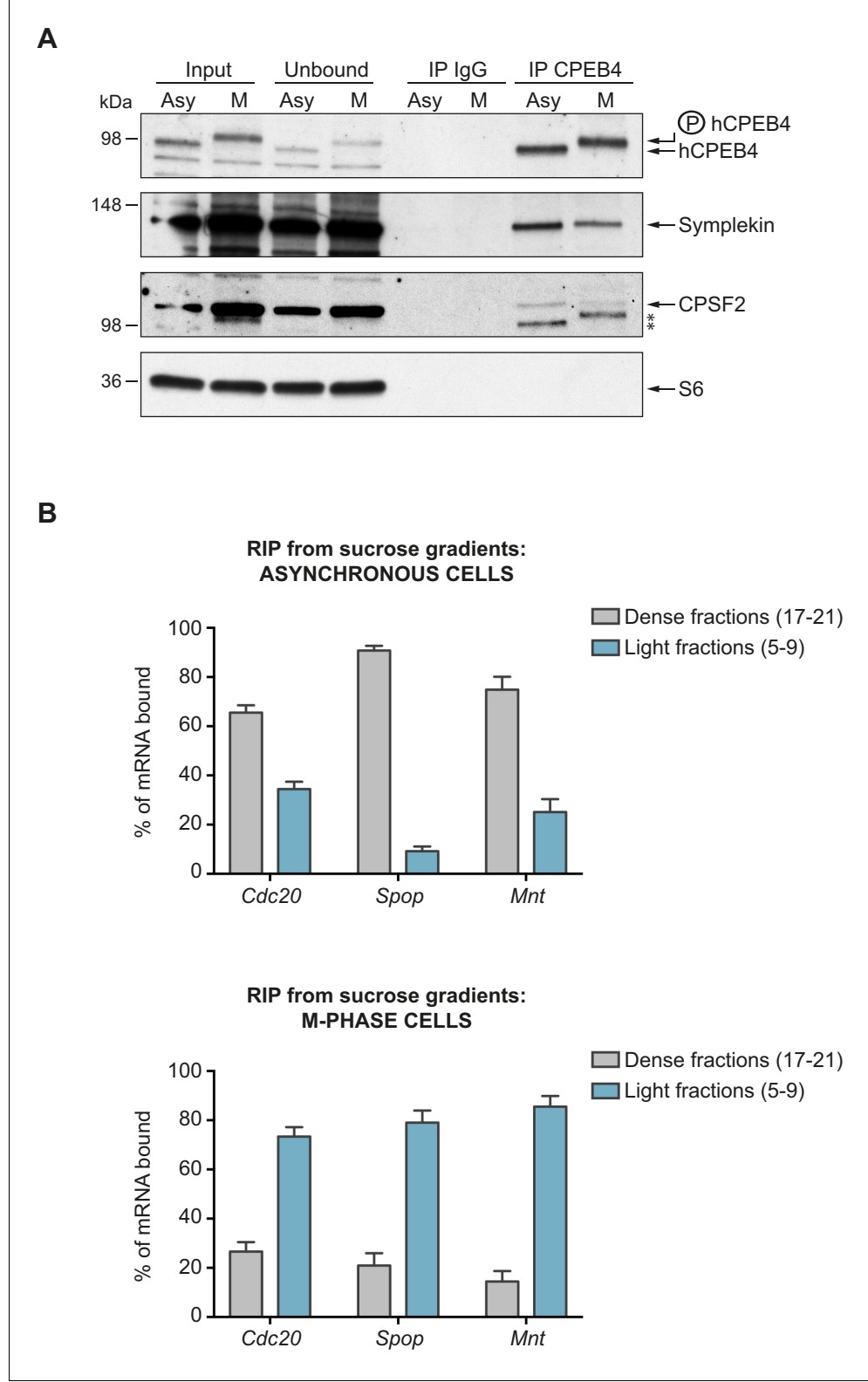

**Figure 7.** Phosphorylated/monomeric and non-phosphorylated/phase-separated hCPEB4 equally interact with cofactors and bind target mRNAs. (**A**) Immunoprecipitation of hCPEB4 from asynchronous ('Asy') and M-phase ('M', synchronized with nocodazole) U2OS cells. IgG was used as negative control. The Co-IPs of Symplekin, CPSF2 and the ribosomal protein S6 were analysed. Asterisks indicate unspecific bands. (**B**) RNA-immunoprecipitation (RIP) of hCPEB4 from 5–40% sucrose gradients dense (17–21) and light (5–9) fractions. The

*Figure 7 continued on next page*

*Figure 7 continued*

upper panel shows hCPEB4 from asynchronous cells, while the lower panel shows hCPEB4 from M-phase cells (synchronized with nocodazole). The binding of hCPEB4 to *Cdc20*, *Spop* and *Mnt* target mRNAs was assessed by RT-qPCR. Results are shown as % of mRNA bound by hCPEB4 in each fraction (results were normalized by the negative control *Gapdh*). Two independent biological replicates, each one with three technical replicates, are shown as means and SEM.

multisite phosphorylations is reminiscent of the ultrasensitive responses that drive irreversible, discrete phase transitions in meiosis, as exemplified by the switch-like regulation of Cdc25C by Cdk1 hyperphosphorylation. Partial phosphorylation of half of the possible sites already provides the highest ultrasensitivity, which may explain the polyadenylation and cell cycle progression effects of the 12A CPEB4 mutant, although it is also phosphorylated in unidentified residues (reviewed in ref. [*Ferrell and Ha, 2014*]). Thus, the ultrasensitivity may be driven at least in part by cumulative electrostatic interactions between NTDs (*Borg et al., 2007*).

## Materials and methods

### *Xenopus laevis* oocyte preparation

Stage VI oocytes were obtained from *X. laevis* females as described previously (*de Moor and Richter, 1999*). Maturation was induced with 10 µM progesterone (Sigma-Aldrich, Saint Louis, MO).

### Plasmid constructions

For protein purification, xCPEB4 open reading frame (ORF), full length or fragments, was cloned in pET30a vector downstream of the HIS-tag (*Igea and Méndez, 2010*). For competition and rescue experiments, xCPEB4 ORF and its 3'UTR were cloned in pBSK and an HA-tag was added at the N-terminus (3'UTR nucleotides targeted by the 20AS used for xCPEB4 depletion were not included). The pBSK-Emi2 3'UTR was obtained from (*Belloc and Mendez, 2008*). For cell transfection, xCPEB4 ORF was cloned in pPEU16, which contains an N-terminal OneStrep-eGFP tag, by In-FusionTM (BD Clontech, Mountain View, CA) cloning reaction (*Berrow et al., 2007*).

### Mutagenesis

Mutagenesis of xCPEB4 phosphorylation sites, as well as Y490 and K595 RNA-binding residues (*Afroz et al., 2014*), was performed with the QuickChange Lightning Multi Site-Directed Mutagenesis Kit (Agilent Technologies, Santa Clara, CA), following the manufacturer's instructions and using the following mutagenic primers (mutated codons are highlighted in bold):

 S18A: 5'-CACTGGGAACAAG**GCA**GCTTTTCCAGTC-3'
 S18D: 5'-CACTGGGAACAAG**GAC**GCTTTTCCAGTC-3'
 S38/40A: 5'-CATCATCAGAATACA**GCC**CCC**GCC**CCCGCTGCTTTTATA-3'
 S38/40D: 5'-CAGAATACA**GAC**CCC**GAC**CCCGCTGCTTTTAT-3'
 S97A: 5'-CCTTAGACAAACAGCTC**GCT**CCTAGCCAAAGC-3'
 S97D: 5'-CCTTAGACAAACAGCTC**GAT**CCTAGCCAAAGC-3'
 S250/253A: 5'-CCAACAGCAAAGGAGA**GCT**CCTGCC**GCT**CCCCATCTGCCTCC-3'
 S250/253D: 5'-CCAACAGCAAAGGAGA**GAT**CCTGCC**GAT**CCCCATCTGCCTCC-3'
 S324/330A: 5'- CTTAATGGAGGAATA**GCA**CCCTTGAATTCCATA**GCA**CCATTAAAGAAG-3'
 S328/330A: 5'-CACCCTTGAAT**GCC**ATA**GCA**CCATTAAAG-3'
 S324/328/330D:5'-CTTAATGGAGGAATA**GAT**CCCTTGAAT**GAC**ATA**GAC**CCATTAAAGAAGAAT-3'
 S351A: 5'-GGCAGGCCCAAT**GCT**GCCTTTGCAC-3'
 S351D: 5'-GGCAGGCCCAAT**GAT**GCCTTTGCAC-3'
 S357/362A: 5'-CTTTGCACCAAAA**GCT**TGGATGGATGAC**GCC**TTGAACAGAGC-3'
 S357/362D: 5'-CTTTGCACCAAAA**GAT**TGGATGGATGAC**GAC**TTGAACAGAGC-3'
 Y490A: 5'-CTTTCCTCCGAAAGGT**GCT**GCGTTCTTGCTGTTCC-3'
 K595A: 5'-GAGCTAAAGTACCCC**GCA**GGAGCTGGGCGTG-3'

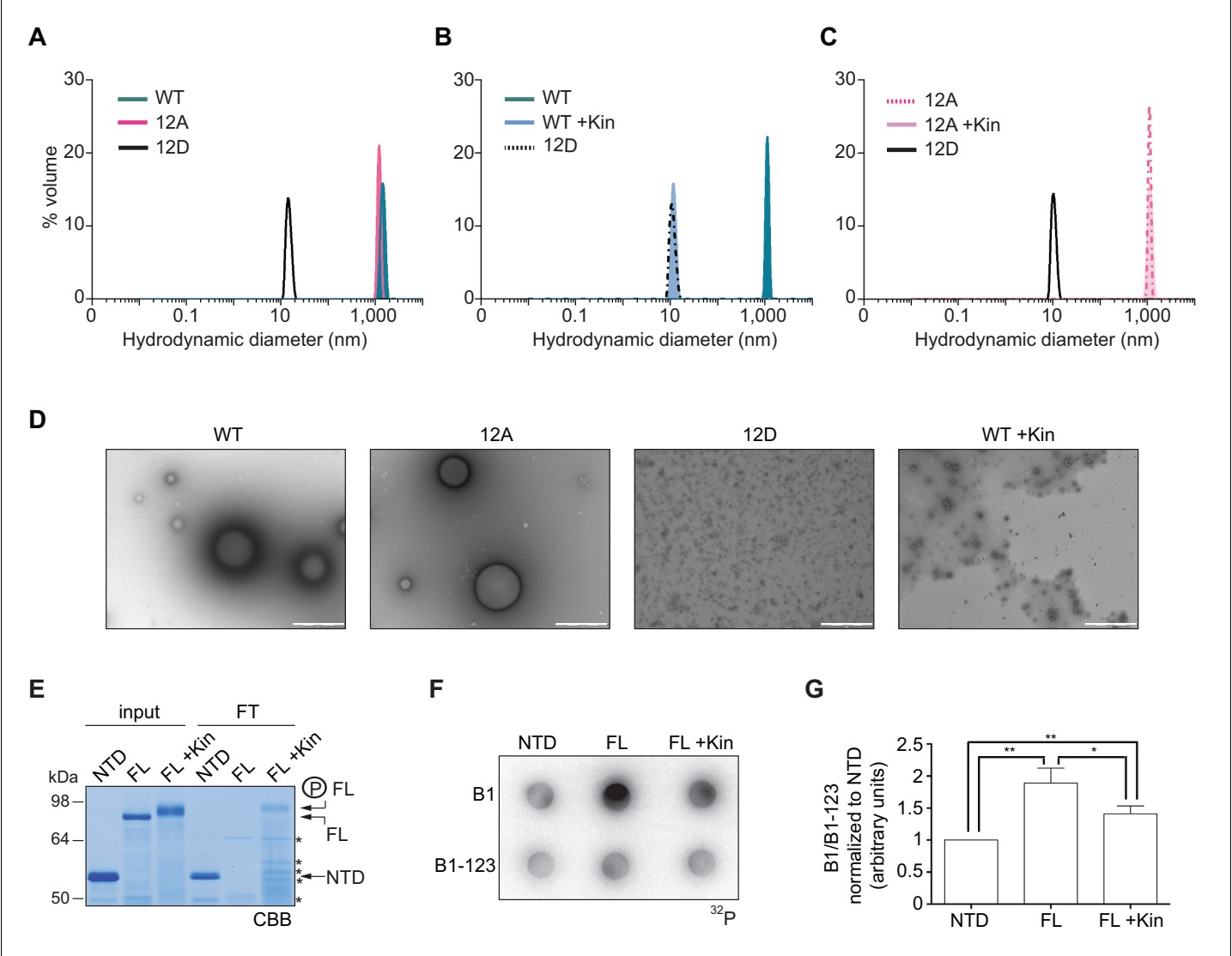

**Figure 8.** Purified xCPEB4 forms liquid-like droplets that are regulated by hyperphosphorylation and that specifically recruit CPE-containing mRNAs. (A) Dynamic light scattering of the xCPEB4 N-terminal domain (NTD), either as wild-type (WT) or the 12A or 12D phosphorylation mutant (with the 12 phosphorylation sites mutated to alanine or aspartic acid, respectively). (B, C) Dynamic light scattering of the xCPEB4 NTD, either as WT (B) or the phospho-null 12A mutant (C), upon phosphorylation (+Kin) with ERK2 and Cdk1/cyclin B. The size distribution by volume is plotted. A representative experiment from three independent replicates is shown. (D) Transmission electron microscopy of the xCPEB4 NTD, either as WT or the 12A or 12D phosphorylation mutant. WT phosphorylated with ERK2 and Cdk1/cyclin B is shown at the right (WT +Kin). Scale bar, 1 μm. (E) Coomassie-stained gel of the input (4%) and the flow-through (FT, 20%) after passing through 0.22-μm-pore-size cellulose acetate filters. NTD, xCPEB4 N-terminal domain; FL, full-length xCPEB4; FL +Kin, FL phosphorylated with ERK2 and Cdk1/cyclin B. Asterisks show degradation products. (F) Cyclin B1–3′-UTR or B1-123–3′-UTR radioactive probe retention on 0.22 μm cellulose acetate filters after incubation with the indicated proteins: NTD, xCPEB4 N-terminal domain; FL, full-length xCPEB4; FL +Kin, FL phosphorylated with ERK2 and Cdk1/cyclin B. (G) Quantification of the cyclin B1–3′-UTR and B1-123–3′-UTR radioactive probe retention from three independent experiments. Results are shown as means and SEM. Significance was determined by the Student's t-test; *p<0.05, **p<0.005.

## Antibodies

Anti-HA (clone 3F10, Sigma-Aldrich), anti-α-Tubulin (clone DM1A, Sigma-Aldrich), anti-Cdk1 (ab18, Abcam, Cambridge, UK), anti-ERK1/ERK2 (9102, Cell Signaling, Danvers, MA), anti-xGLD2 (rabbit polyclonal antibody raised against KRRSDEGNSPYDVKC peptide, *X. laevis*), anti-4E-T (2297, Cell Signaling), anti-DDX6 (PD009, MBL, Nagoya, Japan), anti-hCPEB4 for WB (rabbit polyclonal antibody raised against amino acids 1–302), anti-hCPEB4 for IP (ab83009, Abcam), anti-phospho Histone H3

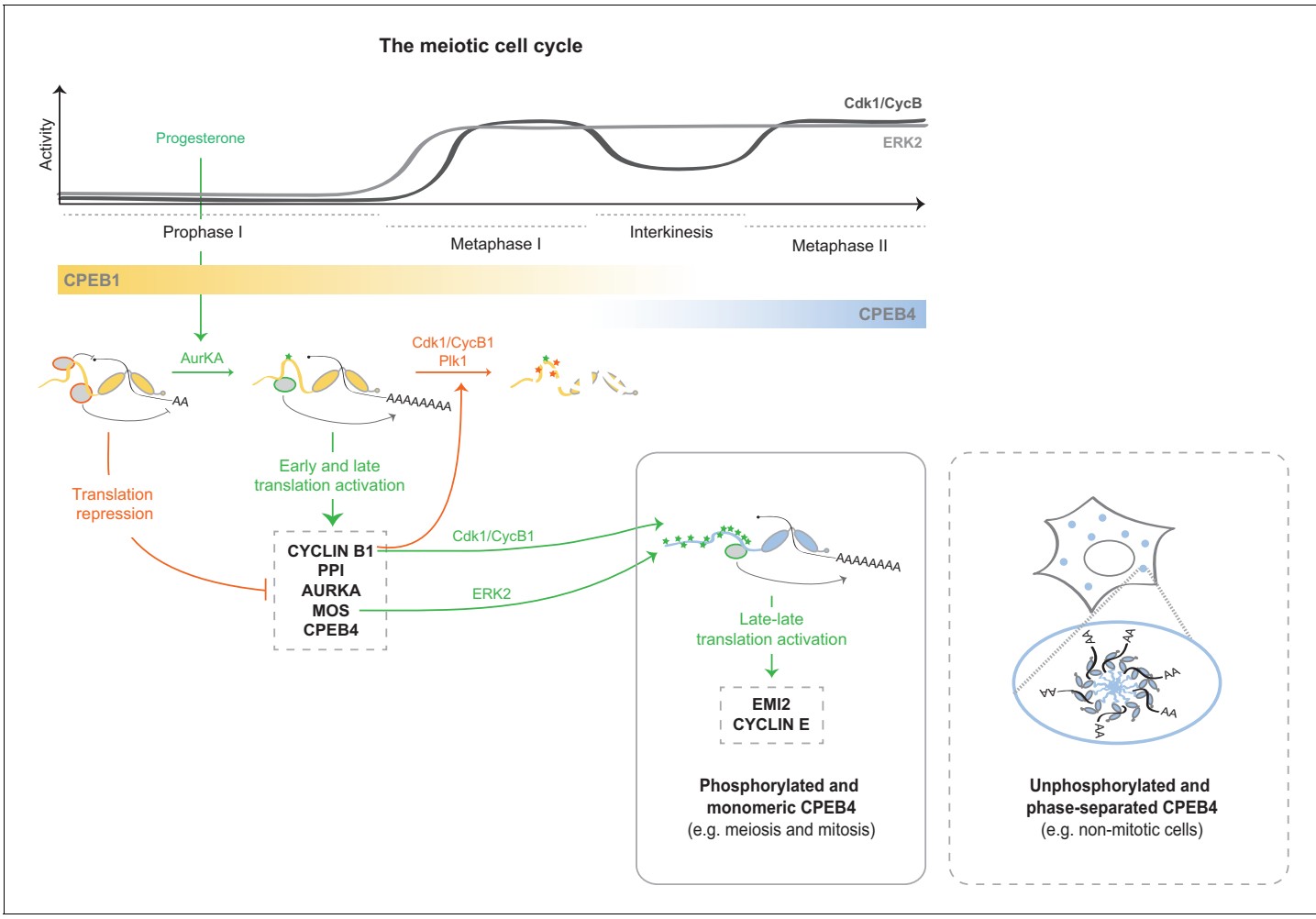

**Figure 9.** Model to illustrate the sequential and coordinated functions of CPEB1 and CPEB4 in the meiotic cell cycle. The different meiotic phases, as well as the activity profiles of ERK2 (light grey) and Cdk1/cyclin B (dark grey) along meiosis, are shown at the top. The sequential expression and function of xCPEB1 (yellow) and xCPEB4 (blue) are shown below. Orange ovals represent CPEB repression complexes. Orange arrows and orange phosphorylation events (*) indicate inhibitory effects. Green ovals represent CPEB activation complexes. Green arrows and green phosphorylation events (*) indicate activating effects. mRNAs activated in the different waves of cytoplasmic polyadenylation are shown in grey boxes. The phase-separation and inactivation that unphosphorylated CPEB4 experiences in non-mitotic cells is represented in the dashed box.

(Ser 10) (06–570, Merck Millipore, Billerica, MA), anti-Symplekin (610644, BD Transduction Laboratories, Franklin Lakes, NJ), anti-CPSF2 (ab126760, Abcam), anti-PARN (ab89831, Abcam), anti-hGLD2 (AP5092, Abgent, San Diego, CA), anti-S6 (2317, Cell Signaling), anti-Vinculin (ab18058, Abcam).

### Lambda protein phosphatase assay (λ-PPase)

Oocytes (injected with 2.43 fmols of *in vitro* transcribed and polyadenylated HA-xCPEB4 RNA) and U2OS cells were lysed with λ-PPase reaction buffer (New England BioLabs, Ipswich, MA) supplemented with 0.4% NP-40 and EDTA-free protease inhibitors (Sigma-Aldrich). λ-PPase (New England BioLabs) reactions were performed following manufacturer's instructions.

### Protein purification

HIS-xCPEB4 (full length or fragments) was expressed in *Escherichia coli*. HIS-tagged proteins from the soluble fraction were purified with $Ni^{2+}$-NTA agarose (Qiagen, Hilden, Germany) in buffer J (25 mM Tris-HCl pH 8, 1 M NaCl, 5 mM $MgCl_2$, 1% NP-40, 5% Glycerol and 6 M Urea) supplemented with 10 mM imidazole pH 8, EDTA-free protease inhibitor cocktail (Sigma-Aldrich) and 1 mM PMSF.

Elution was carried out with elution buffer (25 mM Tris-HCl pH 8, 300 mM NaCl, 5 mM $MgCl_2$, 5% glycerol, 6 M urea, 300 mM imidazole and EDTA-free protease inhibitors). Imidazole and urea concentration were reduced by dialysis.

### In vitro kinase assays

Oocytes were homogenized in H1 kinase buffer (80 mM sodium β-glycerolphosphate pH 7.4, 20 mM EGTA, 15 mM $MgCl_2$, 0.5 mM $Na_3VO_4$) supplemented with 0.5% NP-40 and EDTA-free protease inhibitors (Sigma-Aldrich) (10 µl buffer / oocyte) and centrifuged at 12,000 g for 10 min at 4°C. 1–4 µg of purified recombinant protein (full-length HIS-xCPEB4 or fragments; H1 protein from Sigma-Aldrich) were incubated at RT for 15 min with 8 µl of oocyte extract, 50 µM ATP and 1 µCi of [γ-$^{32}$P] ATP in a final volume of 12 µl. Samples were analysed by SDS-PAGE followed by autoradiography.

For kinase inhibitor assays, 2 µl of metaphase II oocyte extract pre-incubated with specific kinase inhibitors for 30 min at 4°C was used: 33 µM FR180204 (Merck Millipore), 80 µM Roscovitine (Merck Millipore), 0.66 mM SL0101 (Tocris Bioscience, Bristol, UK), 66 nM BI-2536 (BioVision, Milpitas, CA), 130 µM U0126 (Promega, M adison, WI).

For immunodepletion assays, MII oocytes were homogenized in IP lysis buffer (20 mM Tris-HCl pH 8, 0.5% NP-40, 1 mM $MgCl_2$ and 100 mM NaCl) supplemented with H1 kinase buffer and EDTA-free protease inhibitors (Sigma-Aldrich) (30 µl / oocyte). 150 µl of oocyte extracts were incubated with 50 µl of Dynabeads protein A (Invitrogen, Carlsbad, CA) bound to 10 µg IgG (Sigma-Aldrich), 10 µg of anti-Cdk1 or 20 µl of anti-ERK1/ERK2, at 4°C during 20 hr. Immunodepleted oocyte extract from 0.25 oocytes were used for the in vitro kinase assay.

Kinase reactions with ERK2 and Cdk1/cyclin B were performed following the manufacturer's instructions [ERK2 (New England Biolabs), Cdk1/cyclin B (Invitrogen)].

## Mass spectrometry analysis of phosphorylated xCPEB4

Mass spectrometry analysis of xCPEB4 phosphorylation sites generated with MII oocyte extracts was carried out at 'CRG/UPF Proteomics Unit, Centre de Regulació Genòmica (CRG), Universitat Pompeu Fabra (UPF), 08003 Barcelona' as follows: xCPEB4 was phosphorylated in vitro with MII oocyte extracts and re-purified with $Ni^{2+}$-NTA agarose (Qiagen). Samples were in-gel digested with 0.3 µg of trypsin (Promega) (fragment 3) or trypsin followed by chymotrypsin (Roche, Basel, Switzerland) (full length, fragments 1 and 2). After digestion, peptides were extracted and cleaned up on a home-made Empore C18 column (3 M, St. Paul, MN). Samples were analysed using a LTQ-Orbitrap Velos mass spectrometer (Thermo Fisher Scientific, San Jose, CA) coupled to an EasyLC (Thermo Fisher Scientific (Proxeon), Odense, Denmark). Peptides were loaded directly onto the analytical column at 1.5–2 µl/min and were separated by reversed-phase chromatography using a 12 cm column with an inner diameter of 75 µm, packed with 5 µm C18 particles (Nikkyo Technos Co., Ltd. Japan). Chromatographic gradients started at 97% buffer A and 3% buffer B with a flow rate of 300 nl/min, and gradually increased to 93% buffer A and 7% buffer B in 1 min, and to 65% buffer A/35% buffer B in 60 min. Buffer A: 0.1% formic acid in water. Buffer B: 0.1% formic acid in acetonitrile. The mass spectrometer was operated in positive ionization mode with nanospray voltage set at 2.2 kV and source temperature at 250°C. The instrument was operated in DDA mode and full MS scans with 1 micro scans at a resolution of 60,000 were used over a mass range of m/z 250–2000 with detection in the Orbitrap. Auto gain control (AGC) was set to 1E6, dynamic exclusion (60 s) and charge state filtering disqualifying singly charged peptides was activated. In each cycle of DDA analysis, following each survey scan, the top twenty most intense ions with multiple charged ions above a threshold ion count of 5000 were selected for fragmentation at normalized collision energy of 35%. Each sample was injected twice: fragment ion spectra produced via collision-induced dissociation (CID) and via high-energy collision dissociation (HCD). An isolation window of 2.0 m/z, activation time of 0.1 ms and maximum injection time of 100 ms were used. All data were acquired with Xcalibur software v2.2. For data analysis, the Mascot search engine (v2.3, Matrix Science) was used for peptide identification. The data were searched against NCBInr (Xenopus laevis). A precursor ion mass tolerance of 10 ppm at the MS1 level was used. Up to three miscleavages for trypsin were allowed. The fragment ion mass tolerance was set to 0.5 Da. Oxidation of methionine, protein acetylation at the N-terminal, and phosphorylation at serine, threonine and tyrosine were defined as variable modifications;

whereas carbamidomethylation on cysteines was set as a fix modification. Peptides were filtered using an expectation value [EXPECT] cutoff of 0.1.

Mass spectrometry analysis of xCPEB4 phosphorylation sites generated with ERK2 or Cdk1/cyclin B recombinant kinases was done at the Mass Spectrometry Core Facility at the Institute for Research in Biomedicine (IRB Barcelona) as follows: samples were in-gel digested with trypsin at the Proteomics Platform at the Barcelona Science Park. After the digestion, samples were dried in the speedvac and resuspended in 20 µl of 1% formic acid for LC-MS/MS analysis. Digested peptides were diluted in 1% FA. Samples were loaded into a 180 µm × 2 cm C18 Symmetry trap column (Waters Corp., Milford, MA) at a flow rate of 15 µl/min using a nanoAcquity Ultra Performance LCTM chromatographic system (Waters Corp.). Peptides were separated using a C18 analytical column with a 60 min run, comprising three consecutive steps with linear gradients from 1% to 35% B in 60 min, from 35%delete to 50% B in 5 min, and from 50% to 85% B in 3 min, followed by isocratic elution at 85% B in 10 min and stabilization to initial conditions (A = 0.1% FA in water, B = 0.1% FA in $CH_3CN$). The column outlet was directly connected to an Advion TriVersa NanoMate (Advion, Ithaca, NY) fitted on an LTQ-FT Ultra mass spectrometer (Thermo Fisher Scientific). The mass spectrometer was operated in a data-dependent acquisition (DDA) mode. Up to six of the most intense ions per scan were fragmented and detected in the linear ion trap. Target ions already selected for MS/MS were dynamically excluded for 30 s. Spray voltage in the NanoMate source was set to 1.70 kV. The spectrometer was working in positive polarity mode and singly charge state precursors were rejected for fragmentation. For data analysis, a database search was performed with Proteome Discoverer software v1.3 (Thermo Fisher Scientific) using the Sequest search engine and SwissProt database (Xenopodinae, release 2013_06 and the common Repository of Adventitious Proteins, cRAP database). Searches were run against targeted and decoy databases to determine the false discovery rate (FDR). Search parameters included trypsin enzyme specificity, allowing for two missed cleavage sites, carbamidomethyl in cysteine as static modification and methionine oxidation, and phosphorylation at serine, threonine and tyrosine as dynamic modifications. Peptide mass tolerance was 10 ppm and the MS/MS tolerance was 0.8 Da. Peptides with a q-value lower than 0.1 and a FDR < 1% were considered as positive identifications with a high confidence level.

## Two-dimensional phosphopeptide maps

xCPEB4 fragments were phosphorylated *in vitro* as described previously. When using oocyte extracts as the source of kinases, xCPEB4 was re-purified from the kinase reaction with $Ni^{2+}$-NTA agarose (Qiagen). Samples were in-gel digested with sequencing grade modified trypsin (fragment 3) or a combination of trypsin and chymotrypsin (fragment 1 and 2) (both enzymes from Promega). Phosphopeptides were extracted and loaded in glass-backed TLC plates (Sigma-Aldrich or C.B.S. Scientific, Del Mar, CA). The first dimension thin-layer electrophoresis was carried out in a Hunter Thin Layer Peptide Mapping Electrophoresis System (C.B.S. Scientific) with pH 1.9 buffer (2.2% formic acid/7.8% glacial acetic acid in deionized water) at 1000V for 45 min (Sigma-Aldrich TLC plates) or 35 min (C.B.S. Scientific TLC plates). The second dimension thin-layer chromatography was run in Phospho-chromatography buffer (37.5% 1-butanol/25% pyridine/7.5% glacial acetic acid in deionized water). Plates were dried, exposed, and developed with phosphorimager (*Mendez et al., 2000a*).

## Competition experiments

Competition experiments were performed as described previously (*Afroz et al., 2014*) with 0.146 pmols of *in vitro* transcribed and polyadenylated RNA encoding for HA-CPEB4 (WT or mutants). 0% competition was assigned to WT HA-CPEB4, while 100% competition was assigned to the HA-CPEB4-12A mutant. The percentage of competition of the other conditions was normalized on the basis of these values.

## Anti-sense oligonucleotide and phenotypic rescue experiment

Rescue experiments were performed as described previously (*Igea and Méndez, 2010*) with some modifications. In order to ablate xCPEB4 expression, 116 ng of 20AS oligonucleotide (5'-GCAA TGGGTTGCTCAGTTCCA-3') targeting xCPEB4 3'UTR were injected in stage VI oocytes. As a control, we used 23S oligonucleotide (5'-CTTTGCAAGCATCCAAATAAG-3'). After O/N incubation at

18°C, 2.43 fmols of *in vitro* transcribed HA-xCPEB4 + 3'UTR (WT or phospho-mutants) were injected. Oocytes were incubated for 1 hr at RT and maturation was induced. Oocytes were collected 2.5 hr after GVBD, fixed, and stained with Hoechst (20 µg/l). Images were obtained with a Nikon TE200 microscope (x 100 magnification, Olympus DP72 camera).

## Poly(A) tail assay

RNA-ligation-coupled RT-PCR was performed as described previously with some modifications (*Charlesworth et al., 2002*; *Igea and Méndez, 2010*). 4 µg of total oocyte RNA were ligated to 0.5 µg of SP2 anchor primer (5'-P-GGTCACCTCTGATCTGGAAGCGAC-NH2-3') in a 10 µl reaction using T4 RNA ligase (New England Biolabs), following manufacturer's indications. RNA-ligation reaction was reverse transcribed with the RevertAid First Strand cDNA Synthesis Kit (Thermo Fisher Scientific), using 0.5 µg ASP2T antisense primer (5'- GTCGCTTCCAGATCAGAGGTGACCTTTTT-3') in a reaction volume of 50 µl. 0.6 µl of cDNA were used to do gene-specific PCR reactions (30 µl) with BioTaq Polymerase (Bioline, London, UK). The primer used for Emi2 PCR was 5'- GTATATACATTCA TTTGTTCAATGT TGCC-3'. 7.5 µl of PCR were loaded in 1.7% agarose gel and a Southern Blot was performed, using as probe the radiolabeled primer 5'- TGCAGCTAAATAGGTAGACGACATAC-3'.

## Immunoprecipitation from *X. laevis* oocytes

Stage VI oocytes were injected with 4.9 fmols of *in vitro* transcribed HA-xCPEB4 + 3'UTR (wild type or phospho-mutants) and collected at indicated times. Oocytes were homogenized with 10 µl/oocyte of IP lysis buffer (20 mM Tris-HCl pH 8, 1 mM EDTA, 0.5% NP-40, 1 mM $MgCl_2$ and 100 mM NaCl) supplemented with H1 kinase buffer and EDTA-free protease inhibitors (Sigma-Aldrich). 280 µl of oocyte extract was pre-cleared with 25 µl of Dynabeads protein G (Invitrogen) and then incubated for 2 hr at 4°C with anti-HA antibody covalently cross-linked to Dynabeads protein G. Immunoprecipitates were washed six times with IP lysis buffer and eluted with Laemmli sample buffer by heating at 65°C for 20 min. Eluates were resolved in SDS-PAGE, and analysed by Western blotting, silver staining (Pierce Silver Stain for Mass Spectrometry, Thermo Fisher Scientific) or colloidal blue staining (Invitrogen). Mass spectrometry analysis was performed at the Mass Spectrometry Core Facility at IRB Barcelona, as described previously.

## Cell culture, DNA transfection and cell synchronization

U2OS cells were grown in Dulbecco's Modified Eagle Medium (DMEM) supplemented with 10% fetal bovine serum (FBS) and 1% penicillin/streptomycin. When specified, cells were transfected at 80% confluence with 2.5 µg of pPEU16_xCPEB4 plasmid (full length or fragments; WT or phospho-mutants) using Lipofectamine LTX and PLUS Reagent (Life Technologies, Carlsbad, CA). For selection of M-phase cells, cells were synchronized at M-phase with Nocodazole (Sigma-Aldrich) at 100 ng/µl for 16 hr. Alternatively, cells were synchronized at the G2/M with the Cdk1 inhibitor RO-3306 (Merck Millipore) at 9 µM for 21 hr, released and collected 45 min after release (M-phase). In both cases, mitotic cells were collected by mitotic shake-off.

## GFP-xCPEB4 distribution in U2OS cells

Cells were plated and transfected on glass coverslips. 24 hr after transfection, cells were fixed with 1X PBS/4% formaldehyde (Electron Microscopy Sciences, Hatfield, PA) for 10 min at RT, permeabilized with 1X PBS/0.2% Triton X-100 for 5 min at RT and mounted with VECTASHIELD Mounting Medium with DAPI (Vector Laboratories, Burlingame, CA). Image acquisition was performed with a High Throughput Automated Wide-field Olympus IX81 Microscope (Olympus Life Science Europe, Waltham, MA) and a 20x objective. ScanR Acquisition software was used to automatically take 64 images per condition. 100 cells from each condition were analysed with Fiji software.

## Fluorescence recovery after photobleaching (FRAP)

Fluorescence Recovery After Photobleaching was preformed on a Spinning Disk Microscope (Andor Revolution xD, Andor, Belfast, Ireland) equipped with a FRAPPA module allowing fast switching between FRAP and imaging with the same 488 nm laser set at about 5–10 mW. Typical frame rate achieved was 15 images per second with an exposure time of 50 ms on an EMCCD camera (Andor). Image sequences were analysed with Fiji (ImageJ) to register and quantify intensity. Exported data

was analysed in Igor (Wavemetrics) with a custom FRAP procedure (open source code version by kota miura. (2016). FrapCalc: First github release. Zenodo. 10.5281/zenodo.46873; https://zenodo.org/record/46873). FRAP curves were fitted with a single exponential model and half time and mobile fractions were extracted.

## Immunofluorescence

Cells were plated, transfected, fixed and permeabilized as described. Blocking was performed with 1X PBS/10% FBS/0.03% Triton X-100 for 1 hr at RT. Incubation with primary antibodies was done O/N at 4°C, followed by three 5 min washes with 1X PBS. Finally, samples were incubated with the corresponding secondary antibodies for 1 hr at RT, washed three times with 1X PBS and mounted with VECTASHIELD Mounting Medium with DAPI (Vector Laboratories). Images were obtained on an inverted Leica TCS SP5 confocal microscope with a 63x/1.40–0.60 Oil objective.

## Sucrose gradients

U2OS cells were cultured and synchronized at M-phase when specified. Cells were washed with 1X PBS and crosslinked with 0.5% formaldehyde in 1X PBS for 5 min at RT. Crosslinking was stopped with 0.25 M glycine during 4 min. After, cells were washed twice with cold 1X PBS, frozen at −80°C for 10 min and lysed with RIPA buffer (50 mM Tris-HCl pH 8, 150 mM NaCl, 1 mM $MgCl_2$, 1 % NP-40, 0.5% sodium deoxicholate, 0.1% SDS, 1 mM EDTA) supplemented with H1K buffer 1X (80 mM sodium β-glycerolphosphate pH 7.4, 20 mM EGTA, 15 mM $MgCl_2$, 0.5 mM $Na_3VO4$), EDTA-free protease inhibitor cocktail (Sigma-Aldrich), phosphatase inhibitor cocktail 3 (Sigma-Aldrich) and Ribo-Lock RNAse Inhibitor (Thermo Fisher Scientific). Cell lysate was sonicated 15 min at low intensity and centrifuged 30 min at 12000 g and 4°C. The supernatant was recovered. 200 μg of cellular extract were loaded in 5%–40% sucrose gradient and centrifuged at 50000 rpm with MLS-50 rotor (Beckman Coulter, Brea, CA) for 4 hr at 4°C. Fractions of 200 μl were manually obtained from the top to the bottom of the tube. Each fraction was subjected to protein precipitation with trichloroacetic acid, resolved in SDS-PAGE and analysed by Western blotting.

## Immunoprecipitation from U2OS cell extracts

Cellular extracts were prepared as described in the previous section. Cellular extract (800 μg) was pre-cleared with 20 μl of Dynabeads protein A (Invitrogen) for 30 min at 4°C. After, it was incubated with 10 μg of anti-hCPEB4 antibody covalently cross-linked to 50 μl of Dynabeads protein A, for 4 hr at 4°C. Immunoprecipitates were washed five times with RIPA buffer, eluted with Laemmli sample buffer by heating at 65°C for 20 min, resolved in SDS-PAGE and analysed by Western blotting.

## RNA-immunoprecipitation RT-qPCR from sucrose gradients

Fractions obtained from sucrose gradients were pooled as DENSE FRACTIONS (17+19+21) or LIGHT FRACTIONS (5+7+9) and diluted 1:1 in RIPA buffer supplemented with EDTA-free protease inhibitor cocktail (Sigma-Aldrich), phosphatase inhibitor cocktail 3 (Sigma-Aldrich) and RiboLock RNAse Inhibitor (Thermo Fisher Scientific). The resulting extract was pre-cleared with 20 μl of Dynabeads protein A (Invitrogen) for 30 min at 4°C. After, it was incubated with 10 μg of anti-hCPEB4 antibody covalently cross-linked to 50 μl of Dynabeads protein A, for 4 hr at 4°C. Immunoprecipitates were washed five times with RIPA buffer. 2/3 of the immunoprecipitate were digested with Proteinase K during 1 hr at 65°C, and the RNA was isolated by phenol-chloroform extraction. All the RNA was used for reverse transcription with random hexamers and Oligo d(T)$_{20}$, using SuperScript IV Reverse Transcriptase (Invitrogen) and following manufacturer's instructions. The resulting cDNA was used for gene-specific qPCR, using the following primers:

*Cdc20* FW: 5'- GCAAGCTCTGGTGACATCCT-3'
*Cdc20* RV: 5'- ACATGGTGTTCTGCTACCCG-3'
*Spop* FW: 5'-GAGGGGAAGAGACTGCATTG-3'
*Spop* RV: 5'-GCAGCAACAGGGTTTTCATT-3'
*Mnt* FW: 5'-TAGTGGCTGTTCATGCTCCA-3'
*Mnt* RV: 5'-TCTCAGGTTTCAGTGCAGGGTT-3'
*Gapdh* FW: 5'- CGCTCTCTGCTCCTCCTGTT-3'
*Gapdh* RV: 5'- CCATGGTGTCTGAGCGATGT-3'

### Dynamic light scattering (DLS)

Purified xCPEB4 N-terminal domain, in 25 mM Tris-HCl pH 8, 100 mM NaCl, 5% glycerol, 5 mM $MgCl_2$ and 2 M urea buffer, was diluted in a non-containing urea buffer to a final protein concentration of 0.5 mg/ml and 0.2 M urea (other buffer components were kept at the same concentration). Samples were incubated at 4°C for 30 min before DLS analysis. For phosphorylation assays, proteins were diluted in NEBuffer for PK (New England Biolabs) containing 200 µM ATP, incubated on ice for 30 min and immediately phosphorylated with Cdk1/cyclin B (20 ng / µg xCPEB4 N-terminal domain) and ERK2 (10 U / µg xCPEB4 N-terminal domain) at 30°C for 30 min. Kinase reactions were stopped with 20 mM EDTA and kept at 4°C. 100 µl of sample were analysed by DLS with a Zetasizer Nano-S instrument (Malvern, Malvern, UK) at 25°C. Three measurements were obtained for each condition. Three independent experiments were performed.

### Transmission electron microscopy (TEM)

Sample preparation was performed as described for DLS. 6 µl of sample were deposited on carbon film only grids (CF200-CU, Electron Microscopy Sciences) and stained with uranyl formate for 1 min. EM was performed with a Tecnai Spirit microscope (EM) (FEI, Eindhoven, The Netherlands), equipped with a LaB6 cathode. Images were acquired at 120 kV and RT with a 1376 × 1024 pixel CCD camera (FEI, Eindhoven, The Netherlands).

### *In vitro* xCPEB4 liquid-like droplets RNA-binding assay

4 µl of purified xCPEB4, full-length or N-terminal domain, at 2 mg/ml and 6 M urea were diluted with a urea-free buffer to a final volume of 12 µl in the presence of 3 µg of radiolabelled B1 or B1-123 RNA probe and 20 µg of tRNA. Samples were incubated 15 min at RT. After, proteins were diluted to a final volume of 100 µl in NEBuffer for PK (New England Biolabs) containing 200 µM ATP, incubated 30 min at 4°C and, when specified, phosphorylated with 500 U ERK2 and 1 µg Cdk1/cyclin B for 30 min at 30°C. Following phosphorylation, samples were loaded in Corning Costar Spin-X cellulose acetate 0.22 µm tube filters and centrifuged at 12,000 *g* for 1 min. Flow through was recovered and the filter column was washed with NEBuffer for PK. 4% of input and 20% of flow through were resolved in SDS-PAGE. Filters were exposed and developed with phosphorimager.

## Acknowledgements

We thank the Advance Digital Microscopy and Mass Spectrometry Facilities at IRB Barcelona, the CRG/UPF Proteomics Unit and the Electron Cryomicroscopy service at CCiT UB, for technical advice and support. We also thank Laure Weill, Eulàlia Belloc and members of R Méndez laboratory for useful discussions, as well as F Gebauer and J Díez laboratories for scientific feedback and reagents. IRB Barcelona is the recipient of a Severo Ochoa Award of Excellence from MINECO (Government of Spain).

## Additional information

### Funding

| Funder | Grant reference number | Author |
|---|---|---|
| Spanish Government | BFU2011-30121 | Raúl Méndez |
| Spanish Government | Consolider RNAREG CSD2009-00080 | Raúl Méndez |
| Spanish Government | AP2010-1703 | Jordina Guillén-Boixet |
| Spanish Government | BIO2012-31043 | Víctor Buzon Xavier Salvatella |
| Fundación Botín from Banco santander | | Raúl Méndez |
| Asociación Española Contra el Cáncer | | Raúl Méndez |

| Worldwide Cancer Research | 16-0026 | Raúl Méndez |

The funders had no role in study design, data collection and interpretation, or the decision to submit the work for publication.

## Author contributions

JG-B, Performed all the experiments shown in main and supplementary figures, Conception and design, Analysis and interpretation of data, Drafting or revising the article; VB, Contributed to the design of the approach and the development of the techniques used in Figure 8, Analysis and interpretation of data, Drafting or revising the article; XS, RM, Direction of the study, Conception and design, Analysis and interpretation of data, Drafting or revising the article

## Author ORCIDs

Xavier Salvatella, http://orcid.org/0000-0002-8371-4185
Raúl Méndez, http://orcid.org/0000-0002-1952-6905

## Ethics

Animal experimentation: This study was performed in strict accordance with the recommendations of the Ethics Committee from the PCB (CEEA-PCB). The protocol used was approved by the Ethics Committee from the PCB (CEEA-PCB, 5990). Every effort was made to minimize suffering.

## Additional files

### Supplementary files

• Supplementary file 1. xCPEB4 12A- or 12D-interacting proteins. HA-xCPEB4 12A or 12D were immunoprecipitated, and the interacting proteins were analysed by mass spectrometry (relative to *Figure 3—figure supplement 4*). Only proteins with a fold enrichment > 1.5 with respect to the control IP are shown. Two independent experiments were performed.

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
