## [Decision Letter]

Thank you for submitting your article "CPEB4 is regulated during cell cycle by ERK2/Cdk1-mediated phosphorylation and its assembly into liquid-like droplets" for consideration by *eLife*. Your article has been reviewed by two peer reviewers, one of whom, Nick Proudfoot (Reviewer #1), is a member of our Board of Reviewing Editors and the evaluation has been overseen by Tony Hunter as the Senior Editor.

The reviewers have discussed the reviews with one another and the Reviewing Editor has drafted this decision to help you prepare a revised submission.

This study describes the discovery and role of multiple phosphorylations on the regulatory RNA binding protein CPEB4 in both *Xenopus* oocyte development and in a human cell line, U2OS. The study begins by showing that the N-terminal half of this protein (NTD) is subject to multiple phosphorylation events (Figure 1 and Figure 2). Both the sites of phosphorylation as well as the likely kinases involved (Erk2 and Cdk1) are identified by using specific kinase inhibitors together with biochemistry and mass spec analysis. The study goes on to show that phosphorylation of these specific residues are required for efficient polyadenylation activation. The final and arguably most important remaining parts of this study show that CPEB4 NTD when unphosphorylated coalesces into large structures/particles visualised as cytoplasmic bodies. These are detected by transfecting GFP-tagged CPEB4 constructs into human U2OS cells or as liquid droplets characterized by sucrose gradient fractionation and *in vitro* EM studies.

Overall, the reviewers find this study interesting and topical. However we have identified several points that need additional experimentation. Also reviewer 2 has provided a list of more detailed points that should also be addressed in the revision.

Essential revisions:

1) It would be valuable to back up the kinase chemical inhibition experiments by specifically depleting Erk2 and Cdk1 to confirm their role in CPEB4 phosphorylation. Presumably this can be done using antisense RNAs – successfully used in later experiments to deplete CPEB4.

2) In these studies exogenously expressed xCPEB4 protein mutants (dominant negative) were employed to compete with endogenous xCPEB4 in activating synthetic substrate RNA polyadenylation. These data provide evidence that xCPEB4 NTD phosphorylation is required for polyadenylation. Even though this competition assay has been employed in previous studies, it still appears to be a somewhat indirect approach to measure polyadenylation. It would be desirable to test the critical xCPEB4 protein mutants in oocytes depleted for endogenous xCPEB4 so that a direct measurement of their effects on substrate RNA polyadenylation can be tested.

3) A major issue is that it is assumed that the inactivation of the CPEB4 is due to the formation of liquid droplets that may sequester the mRNA targets. However, this conclusion has not been directly addressed. By definition the protein in liquid droplets is in equilibrium with the cytoplasmic soluble pool and therefore the reason why the protein is inactive could be indirect. For example, the protein fails to recruit some protein partner or cofactor. Liquid droplet formation may be a consequence not the cause of the inactivity. Also, it is not clear what fraction of the mRNA pool is sequestered in droplets and whether this can explain why the poly(A) tails are not extended. Therefore, this value and the proportion of the total CPEB4 population that is in the liquid droplet state need to be directly measured. In addition, are the interactions with known CPEB4 binding partners affected, as determined, for example, using co-immunoprecipitation with the human protein in human cells? This has been partially addressed for a subset of binding partners. But whether PARN or GLD2 colocalize together with CPEB4 in the droplets has not been investigated and should be. If the interactions are not affected as the authors claim, these proteins should also be present in the droplets and this raises the question of why the complexes are inactive. Such an outcome would need to be discussed in the paper.

---

## [Author Response]

*[…] Essential revisions:*

*1) It would be valuable to back up the kinase chemical inhibition experiments by specifically depleting Erk2 and Cdk1 to confirm their role in CPEB4 phosphorylation. Presumably this can be done using antisense RNAs – successfully used in later experiments to deplete CPEB4.*

The identification of the kinases that phosphorylate CPEB4 is based in three evidences. As the reviewer points, each approach has its own limitations, but together we believe make a strong argument for the identity of the kinases. (i) The time course of xCPEB4 (FL and fragments) phosphorylation shows that xCPEB4 is not phosphorylated in Prophase I (PI) but only after metaphase I (MI), moreover one of the fragments follows the well-characterized, and unique in oocytes, profile of Cdk1. Based on the very numerous studies dealing with kinase-activities during meiosis, these profiles narrow down the suspected kinases to a very limited number. (ii) Then, we test these candidates with inhibitors. (iii) We phosphorylate recombinant xCPEB4 with purified/recombinant kinases and compare the resulting phosphopeptide maps with those obtained with phase-specific oocyte extracts.

Following the reviewers’ suggestion, we have now added another approach to verify the identity of the kinases. However, mRNA-targeting (to deplete the encoded protein) has two relevant technical and conceptual limitations in this case. First, in *X. laevis* oocytes, this approach is only feasible for maternally encoded proteins (like xCPEB4). For proteins that are already present in PI-arrested oocytes (like ERK2 and Cdk1), targeting their mRNAs does no result in reduction of the proteins within the timeframe of the experiments. The second one, and this would apply also to mRNA-targeting in cell cultures, inhibition of ERK2 and Cdk1 in cells/oocytes will block the entry of the cell/oocyte into Metaphase. Because CPEB4 is only phosphorylated in Metaphase, the result of the experiment will be similar to the above-mentioned time-course. It will say that CPEB4 kinases are activated in metaphase but not whether CPEB4 is a direct or indirect target of ERK2 and Cdk1. This is the reason why, in the kinase-inhibitor experiments, first we obtain oocyte extracts synchronized for each meiotic-phase and then treat the extracts with the inhibitors. This approach, with the intrinsic limitations of the specificity of the inhibitors, points to a direct phosphorylation of CPEB4 by the targeted kinase(s).

To complement these approaches, we have now immunodepleted ERK2 and Cdk1 from oocyte extracts synchronized in metaphase II (New Figure 2—figure supplement 2). Cdk1 immunodepletion leads to a 21% reduction of xCPEB4 phosphorylation (note that Cdk1 is responsible for the phosphorylation of 5 out of the 12 identified phosphorylation sites). In the case of ERK2, which phosphorylates at least 7 residues on xCPEB4 NTD, its immunodepletion results in a 26% further reduction of xCPEB4 phosphorylation, even though ERK2 depletion was not complete. Although it is almost impossible to immunodeplete two kinases to get a near-complete abolishment of their activities, we have found a significant reduction, but not complete inhibition, of xCPEB4 phosphorylation.

*2) In these studies exogenously expressed xCPEB4 protein mutants (dominant negative) were employed to compete with endogenous xCPEB4 in activating synthetic substrate RNA polyadenylation. These data provide evidence that xCPEB4 NTD phosphorylation is required for polyadenylation. Even though this competition assay has been employed in previous studies, it still appears to be a somewhat indirect approach to measure polyadenylation. It would be desirable to test the critical xCPEB4 protein mutants in oocytes depleted for endogenous xCPEB4 so that a direct measurement of their effects on substrate RNA polyadenylation can be tested.*

That was the rationale behind the meiotic progression rescue experiment performed in the original submission (Figure 3), where endogenous xCPEB4 was replaced by xCPEB4 phospho-mutants 12D or 12A and the progression into Metaphase II was assessed (as shown by the extrusion of the polar body and the assembly of the MII metaphase plate). Because transition from MI to MII and MII arrest requires xCPEB4-mediated polyadenylation of *Emi2* mRNA (Belloc 2008 and Igea 2010), this was an indirect measurement of xCPEB4 activity.

Now we have repeated the experiment, depleting endogenous xCPEB4, expressing xCPEB4 12D or 12A mutants and measuring directly endogenous *Emi2* mRNA polyadenylation (New Figure 3). The result confirms the competition assay in polyadenylation of a reporter and the rescue of meiotic progression, showing that xCPEB4 12D, but not xCPEB4 12A, rescues *Emi2* mRNA polyadenylation.

*3) A major issue is that it is assumed that the inactivation of the CPEB4 is due to the formation of liquid droplets that may sequester the mRNA targets. However, this conclusion has not been directly addressed. By definition the protein in liquid droplets is in equilibrium with the cytoplasmic soluble pool and therefore the reason why the protein is inactive could be indirect. For example, the protein fails to recruit some protein partner or cofactor. Liquid droplet formation may be a consequence not the cause of the inactivity.*

We agree with the reviewers that the cause/consequence relationship between phase-separation and translational inhibition is a difficult issue to be addressed directly (just looking at the P-body literature, for many years, gives an idea of the complexity).

In the case of CPEB4-phosphorylation/phase-separation, we think that a causal effect is more probable based in our observations. Thus, CPEB4 12A is not active in cytoplasmic polyadenylation and mediates the assembly of liquid-like droplets (LLDs), whereas CPEB4 12D drives polyadenylation and translational activation and prevents LLDs. These observations are consistent with both models:

(i)”Consequence”: CPEB4 12A assembles a repression complex, by recruiting specific cofactors, and then this complex is “stored” in LLDs.

(ii) “Cause”: CPEB4 12A, bound to mRNAs, assembles LLDs, which in turn prevent the translational activation (i.e. polyadenylation and ribosomal recruitment).

However, the first model implies that CPEB4 12A should specifically recruit cofactors that mediate the translational repression. Proteomic analysis of the proteins bound to CPEB4 12A and 12D did not identify such cofactors. Moreover, this model would imply that the LLDs would only be generated after the repression complex has been assembled, through the recruitment of co-factors. But, not only the LLDs are generated in the absence of the RRMs required to assemble the repression complex, but also purified recombinant CPEB4 (or CPEB4 N-terminal IDRs) forms LLDs *in vitro*, regulated by phosphorylation, in the absence of co-factors. Thus, the formation of LLDs is an intrinsic property of CPEB4 IDRs and, when the RRMs are present, CPEB4 recruits mRNAs to these LLDs, sequestering the mRNA.

As the reviewer mentions, LLDs and soluble CPEB4 are in equilibrium, which is displaced toward the LLD state when CPEB4 is not phosphorylated, and even further when the LLDs contain mRNA (as shown by FRAP experiments). Thus, in an equilibrium between translation and repression, non-phosphorylated CPEB4 would favor the repression state.

We have included the core of these arguments in the revised version of the manuscript.

Also, it is not clear what fraction of the mRNA pool is sequestered in droplets and whether this can explain why the poly(A) tails are not extended. Therefore, this value and the proportion of the total CPEB4 population that is in the liquid droplet state need to be directly measured.

To address these points we have expanded the analyses of CPEB4 phase-separation in somatic cells, comparing asynchronous cells, where CPEB4 is mainly in LLDs, with M-phase arrested cells (please note that in an asynchronous population 5-8% of cells will be in mitosis). CPEB4 complexes from these two populations were resolved by sucrose gradient-centrifugation and the resulting fractions were analyzed for CPEB4 distribution and for CPEB4-associated mRNAs.

New Figure 6 shows the distribution of CPEB4 along the gradient (as % of total intensity). In asynchronous cells, 75% of CPEB4 is in LLD (dense fractions 15-23), while in M-phase 84% is in a monomeric state (light fractions 1-13).

New Figure 7 shows the association of CPEB4 (monomeric and LLD state) to its target mRNAs: *Cdc20*, *Spop* and *Mnt*. These three candidates were chosen because they are specifically polyadenylated and activated in mitosis and in a CPEB4 dependent manner (Novoa et al., 2010). Thence, RNA-immunoprecipitation from specific sucrose gradient fractions shows: (i) In asynchronous cells, where the majority of CPEB4 is in the LLD state, 63% of *Cdc20*, 92% of *Spop* and 78% of *Mnt* mRNAs bound to CPEB4 are bound by CPEB4 in the LLD state (dense fractions). (ii) In M-phase, when CPEB4 LLDs disassemble, 75% of *Cdc20*, 83% of *Spop* and 88% of *Mnt* mRNAs bound to CPEB4 are bound by monomeric CPEB4 (light fractions). These results show that both physical states of CPEB4 are equally able to bind target mRNAs. We conclude that when CPEB4 is not phosphorylated its target mRNAs are majorly bound by the LLDs, while upon phosphorylation these target mRNAs are bound by monomeric CPEB4. Together with the finding that CPEB4-LLDs do not contain ribosomes, these percentages are consistent with the translational repression and activation of the mRNAs encoding these three cell cycle regulators.

*In addition, are the interactions with known CPEB4 binding partners affected, as determined, for example, using co-immunoprecipitation with the human protein in human cells? This has been partially addressed for a subset of binding partners. But whether PARN or GLD2 colocalize together with CPEB4 in the droplets has not been investigated and should be. If the interactions are not affected as the authors claim, these proteins should also be present in the droplets and this raises the question of why the complexes are inactive. Such an outcome would need to be discussed in the paper.*

Please, note that there are no known “repression specific” co-factors for CPEB4. Identification of such repressor complex was the original aim of the CPEB4 12A IP followed by mass spectrometry, with the conclusion that no such co-factor was detected. Moreover, the association with the poly(A) polymerase GLD2 was unaffected by the phosphorylation state of CPEB4 in oocytes. These results drove us to explore if, unlike CPEB1, where the phosphorylation of a single site changes the identity of the associated proteins, for CPEB4 a change of its physical state itself and not a specific cofactor could be the cause of the regulation of its activity.

Now, we have further analyzed possible CPEB4 co-factors shared with CPEB1 repression/activation complexes. To this aim, we have performed CPEB4 co-IP in mammalian cells followed by western blot for known CPEB1 co-factors and we have checked the distribution of these co-factors along sucrose gradients. We found that both CPSF2 and Symplekin, which are distributed in both dense and light fractions of the gradients, are bound to CPEB4, without differences between the free and the LLD state (New Figure 6 and Figure 7). The association with GLD2 in mammalian cells was difficult to address by co-IP (it co-migrates with the heavy chain of the antibodies used in the IP). Nevertheless, we show that GLD2 is present in both dense and light fractions of the gradients (New Figure 6), which fits with the result obtained in oocytes where we show that GLD2 is equally associated with the 12A (LLD) and the 12D (monomeric) CPEB4 mutants. We failed to detect PARN in the co-IPs and, in fact, PARN is not present in the dense fractions of the gradient (New Figure 6), suggesting that it does not co-localize with CPEB4 in the LLD. We did also look for the presence of ribosomes in the LLD (S6) and we found that there are no ribosomes associated with CPEB4 (New Figure 7).

Altogether, we could not find any co-factor of CPEB4 associating differently depending on its phosphorylation/physical state. However, GLD2, Symplekin and CPSF are present in both CPEB1 repression/deadenylation and activation/polyadenylation complexes (Barnard 2004. Kim 2006). So it seems difficult to extract any functional conclusion from these associations. In conclusion, we believe that our data support the idea that it is the phosphorylation-regulated phase-separation itself, and not the differential binding to positive or negative co-factors, what defines the dual role of CPEB4 as a translational repressor or activator. We have included this argument in the manuscript.